# It's All Connected: A Journey Through Test-Time Memorization, Attentional Bias, Retention, and Online Optimization

**Ali Behrouz** [1]     **Meisam Razaviyayn** [1]     **Peilin Zhong** [1]     **Vahab Mirrokni** [1]

[1] Google Research

`{alibehrouz, razaviyayn, mirrokni}@google.com`

`peilin.zhong@columbia.edu`

## Abstract

This paper introduces MIRAS, a unified framework that reconceives neural architectures (such as Transformers and modern linear RNNs) as associative-memory modules governed by online optimization. In MIRAS, each module learns key-value mappings via an *attentional bias* (the internal learning objective) and preserves past information via a *retention function* (the memory regularizer). This perspective provides a principled reinterpretation of "forgetting" mechanisms as forms of regularization. Our framework reveals a critical limitation: virtually all existing sequence models, including recent unification efforts, are constrained by dot-product similarity or $\ell_2$ loss. MIRAS moves beyond this narrow focus, providing a generative framework that unlocks a richer design space informed by principles from (robust) optimization and statistics. We introduce diverse alternatives—such as $\ell_p$ norms, Huber loss, KL-based losses, and $f$-divergence measures—leading to novel architectural designs with improved stability and robustness. Utilizing this expanded space, we build three novel, attention-free, and parallelizable models (MONETA, MEMORA, YAAD) that combine expressive MLP memories with these new mechanisms. Empirically, different MIRAS instantiations trade off complementary strengths, illustrating the framework's capacity to navigate architectural design choices. Several variants achieve strong scaling, larger effective context windows, and demonstrate results better than state-of-the-art linear RNNs across various tasks, including language modeling, commonsense reasoning, and challenging long-context recall.

## 1 Introduction

Designing efficient architectural backbones for sequence modeling is vital for strengthening foundation models across diverse domains and data modalities such as language (Vaswani et al., 2017; Gemma-Team et al., 2024), computer vision (Dosovitskiy et al., 2020), computational biology (Wang et al., 2024), and neuroscience (Behrouz et al., 2024). Transformers (Vaswani et al., 2017) remain state of the art thanks to their in-context learning and scalability (Kaplan et al., 2020), but their quadratic time and space complexity limits use in long-context applications (Dalal et al., 2025; Liu et al., 2024b; Li et al., 2024).

Recent work tackles Transformers' long-context limits by creating efficient recurrent alternatives (Schlag et al., 2021; Smith et al., 2022). Unlike the Transformer's linearly growing KV cache, these models compress context into fixed-size memory, requiring better memory management for strong performance. To design more effective architectures, studies improves memory capabilities through: (1) richer learning rules, from Hebbian (Hebb, 2005) to Delta (Schlag et al., 2021) and Omega (Behrouz et al., 2025a); (2) advanced forget gates, from LSTM (Schmidhuber & Hochreiter, 1997) to Mamba2 (Dao & Gu, 2024) and RWKV-7 (Peng et al., 2025a); (3) more expressive memory, from vector memory in RetNet (Sun et al., 2023) and LRU (Orvieto et al., 2023) to deep neural memory in Titans (Behrouz et al., 2025c).

Table 1: Overview of recent architectures in MIRAS framework perspective. Surprisingly, all models are using the same type of attentional bias and regularization (forget gate). Note that these architectural choices does not uniquely identify the backbone as there are other design choices (e.g., input-dependency, channel-wise parameters, etc.) as well as the use of other components.

| Model | Memory Architecture | Attentional Bias | Retention Gate | Memory Algorithm | Memory Write Operation |
|---|---|---|---|---|---|
| Shallow Memory | | | | | |
| RetNet (2023) | Vector | Dot-Product | $L_2$ | GD | $\mathcal{M}_t = \alpha\mathcal{M}_{t-1} + \mathbf{v}_t\boldsymbol{k}_t^\top$ |
| Transformer (2017) | Matrix | $L_2$ | - | Nonparametric | $\mathcal{M}_t = \mathcal{M}_{t-1} \cup \{(\boldsymbol{k}_t, \mathbf{v}_t)\}$ |
| LA (2021) | Matrix | Dot-Product | - | GD | $\mathcal{M}_t = \mathcal{M}_{t-1} + \mathbf{v}_t\boldsymbol{k}_t^\top$ |
| DFW | Matrix | Dot-Product | - | GD | $\mathcal{M}_t = (\beta_t\alpha_t^\top)\odot\mathcal{M}_{t-1} + \mathbf{v}_t\boldsymbol{k}_t^\top$ |
| Lightening Attention (2025) | Matrix | Dot-Product | $L_2$ | GD | $\mathcal{M}_t = \alpha\mathcal{M}_{t-1} + \mathbf{v}_t\boldsymbol{k}_t^\top$ |
| GLA (2024b) | Matrix | Dot-Product | $L_2$ | GD | $\mathcal{M}_t = \text{Diag}(\alpha_t)\mathcal{M}_{t-1} + \mathbf{v}_t\boldsymbol{k}_t^\top$ |
| Mamba2 (2024) | Matrix | Dot-Product | $L_2$ | GD | $\mathcal{M}_t = \alpha_t\mathcal{M}_{t-1} + \mathbf{v}_t\boldsymbol{k}_t^\top$ |
| HGRN2 (2024) | Matrix | $L_1$ | $L_2$ | GD | $\mathcal{M}_t = \text{Diag}(\alpha_t)\mathcal{M}_{t-1} + \mathbf{v}_t(1-\alpha_t)^\top$ |
| DeltaNet (2021) | Matrix | $L_2$ | - | GD | $\mathcal{M}_t = \mathcal{M}_{t-1}(\mathbf{I} - \beta_t\boldsymbol{k}_t\boldsymbol{k}_t^\top) + \beta_t\mathbf{v}_t\boldsymbol{k}_t^\top$ |
| Longhorn (2024a) | Matrix | $L_2$ | - | Implicit GD | $\mathcal{M}_t = \mathcal{M}_{t-1}\left(\mathbf{I} - \frac{\beta_t\boldsymbol{k}_t\boldsymbol{k}_t^\top}{1+\beta_t\boldsymbol{k}_t^\top\boldsymbol{k}_t}\right) + \left(\frac{\beta_t}{1+\boldsymbol{k}_t^\top\boldsymbol{k}_t\beta_t}\odot\mathbf{v}_t\right)\boldsymbol{k}_t^\top$ |
| TTT-Linear (2024) | Matrix | $L_2$ | - | GD | $\mathcal{M}_t = \mathcal{M}_{t-1} - \eta\nabla\mathcal{L}(\mathcal{M}_{t-1}, \boldsymbol{k}_t, \mathbf{v}_t)$ |
| Gated DeltaNet (2024a) | Matrix | $L_2$ | $L_2$ | GD | $\mathcal{M}_t = \mathcal{M}_{t-1}(\alpha_t(\mathbf{I} - \beta_t\boldsymbol{k}_t\boldsymbol{k}_t^\top)) + \beta_t\mathbf{v}_t\boldsymbol{k}_t^\top$ |
| RWKV-7 (2025b) | Matrix | $L_2$ | $L_2$ | GD | $\mathcal{M}_t = \mathcal{M}_{t-1}(\text{diag}(\alpha_t) - \beta_t\boldsymbol{k}_t\boldsymbol{k}_t^\top) + \beta_t\mathbf{v}_t\boldsymbol{k}_t^\top$ |
| DeltaProduct (2025) | Matrix | $L_2$ | $L_2$ | MGD* | $\mathcal{M}_t = \mathcal{M}_{t-1}\left(\alpha_t\prod_{i=1}^n(\mathbf{I} - \beta_{t,i}\boldsymbol{k}_{t,i}\boldsymbol{k}_{t,i}^\top)\right) + \sum_{j=1}^n\prod_{i=j}^n(\mathbf{I} - \beta_{t,i}\mathbf{v}_{j,i}\boldsymbol{k}_{j,i}^\top)$ |
| Sliding Window Linear Attention (SWLA) (2025a) | Matrix | Sliding Window Dot-Product | $L_2$ | GD | $\mathcal{M}_t = \alpha_t\mathcal{M}_{t-1} + \sum_{i=t-c+1}^t \gamma_i^t \mathbf{v}_i\boldsymbol{k}_i^\top,$ |
| Deep Memory | | | | | |
| TTT-MLP (2024) | 2-layer MLP | $L_2$ | - | GD | $\mathcal{M}_t = \mathcal{M}_{t-1} - \eta\nabla\mathcal{L}(\mathcal{M}_{t-1}, \boldsymbol{k}_t, \mathbf{v}_t)$ |
| Titans (2025c) | $k$-layer MLP | $L_2$ | $L_2 + L_2^\dagger$ | GD + Momentum | $\mathcal{M}_t = \alpha_t\mathcal{M}_{t-1} + \mathbf{S}_t$ $\mathbf{S}_t = \theta_t\mathbf{S}_{t-1} - \eta_t\nabla\mathcal{L}(\mathcal{M}_{t-1}, \boldsymbol{k}_t, \mathbf{v}_t)$ |
| OmegaNet (2025a) | GLU | Sliding Window $L_2$ | $L_2$ | GD | $\mathcal{M}_t = \alpha_t\mathcal{M}_{t-1} - \sum_{i=t-c+1}^t\gamma_i^t\nabla\mathcal{L}(\mathcal{M}_{i-1}, \boldsymbol{k}_i, \mathbf{v}_i)$ |
| Atlas (2025a) | $k$-layer MLP/GLU | Sliding Window $L_2$ | $L_2$ | Muon | $\mathcal{M}_t = \alpha_t\mathcal{M}_{t-1} + \text{NS-}k(\mathbf{S}_t)$ $\mathbf{S}_t = \theta_t\mathbf{S}_{t-1} - \sum_{i=t-c+1}^t\gamma_i^t\nabla\mathcal{L}(\mathcal{M}_{i-1}, \boldsymbol{k}_i, \mathbf{v}_i)$ |
| LaCT (2026) | GLU | Dot-Product | - | Muon | $\mathcal{M}_t = \mathcal{M}_{t-1} + \text{NS-}k(\mathbf{S}_t)$ $\mathbf{S}_t = \mathbf{S}_{t-1} - \eta_t\nabla\mathcal{L}(\mathcal{M}_{t-1}, \boldsymbol{k}_t, \mathbf{v}_t)$ |
| Hope-Attention / CMS (2025b) | $k$-layer MLP/GLU | NTP‡ | $L_2$ | Arbitrary | $\mathcal{M}_t = \alpha_t\mathcal{M}_{t-1} - \eta\nabla\mathcal{L}(\mathcal{M}_{t-1}, \mathbf{x}_t)$ |
| MONETA (ours) | $k$-layer MLP | $L_p$ | $L_q$ | GD | $A_t = \alpha_t A_{t-1} - \eta_t\nabla\ell(W_{t-1}; \boldsymbol{k}_t, \mathbf{v}_t), W_t = \frac{A_t}{\|A_t\|_q^{q-2}}$ |
| YAAD (ours) | $k$-layer MLP | Huber | $L_2$ | GD | $W_t = \alpha_t W_{t-1} - \begin{cases}\eta_t \nabla\ell_2(W_{t-1}; \boldsymbol{k}_t, \mathbf{v}_t) & \text{if } \|\mathcal{M}(\boldsymbol{k}_t) - \mathbf{v}_t\| \le \delta_t, \\ \eta_t \delta_t\nabla\ell_1(W_{t-1}; \boldsymbol{k}_t, \mathbf{v}_t) & \text{Otherwise.}\end{cases}$ |
| MEMORA (ours) | $k$-layer MLP | $L_2$ | KL | GD | $W_t = \text{Softmax}(\alpha_t\log(W_{t-1}) - \eta_t\nabla\ell(W_{t-1}; \boldsymbol{k}_t, \mathbf{v}_t))$ |

* is using multiple rounds of GD per token. † Titans use local and global retention using $L_2$ loss. ‡ NTP: Next-Token-Prediction.

At the core of these advancements lies a critical question: "what is the underlying design framework behind these sequence models, and how can these models be enhanced?". Taking inspiration from the broad definitions of associative memory and learning in neuropsychology literature (Okano et al., 2000), several studies discuss connections between Transformers and Recurrent Neural Networks (RNNs) with associative memory (Hopfield, 1982; Ramsauer et al., 2021; Bietti et al., 2023). These studies, however, either: (1) lack a *universal* explanation to *fully* illustrate the underlying learning algorithms, (2) are limited to a specific definition of associative memory and lack generalizability, and (3) are unable to describe standard, widely-used components such as forget gate. To address these concerns, several works have tried to unify neural architecture designs. Notably, Liu et al. (2024a) adopted an online learner viewpoint, similar to (Learning-Retaining Viewpoint) in our paper. Concurrently, Wang et al. (2025) adopted an online $L_2$-regression based viewpoint, which connects to the special case of (FTRL Viewpoint) in our work with $L_2$-regression as the objective.

While these frameworks successfully unify existing models, they remain constrained by the $\ell_2$ and dot-product paradigms, effectively making them specific instances of our broader framework. MIRAS distinguishes itself in two critical ways: First, we provide a formal connection between these two viewpoints (Theorem 2.2). Second, and crucially, MIRAS transcends the limitations of Euclidean spaces. Unlike prior frameworks that merely catalog existing $\ell_2$-based methods, MIRAS provides the generative capacity to design novel architectures with enhanced robustness and stability, addressing the sensitivity to outliers inherent in $\ell_2$-based optimization.

**Contributions.** Inspired by the human cognitive phenomenon of attentional bias—the natural tendency to prioritize certain stimuli—we re-examine the foundations of sequence modeling by connecting (online) optimization and associative memory. This perspective allows us to unify existing architectures and unlock a *principled* design space. Our main contributions are as follows:

- *A Unified Framework:* We introduce MIRAS[1], a comprehensive framework that reconceptualizes sequence models (including Transformers and modern RNNs) as associative memory modules governed by online optimization. MIRAS formally defines the core components of these models as Attentional Bias (the internal learning objective) and Retention (the memory regularizer).

---

[1] "Miras" is the translation of "Legacy" in several languages including Persian, Arabic, and Turkish. We choose this name since this framework provides clear steps for future design of sequence models.

- *Theoretical Reinterpretation and Critical Insights:* Through the lens of MIRAS, we provide interpretation of existing forgetting mechanisms (e.g., gates in LSTMs or Mamba2) as specific forms of regularization within online optimization frameworks (e.g., FTRL). Crucially, our unification reveals a significant limitation: virtually all existing successful architectures rely narrowly on $\ell_2$ loss or dot-product similarity for both bias and retention (See Table 1).

- *Expansion of the Architectural Design Space:* MIRAS provides a principled foundation for moving beyond the $\ell_2$ paradigm. We leverage principles from robust optimization and statistics to propose and explore novel, non-Euclidean attentional biases (e.g., Huber loss, $\ell_p$ norms) and retention gates (e.g., KL-divergence, f-divergence), leading to architectures with improved stability and robustness. We specifically derive eight of these *variants* in Section 4.

- *Novel Attention-Free Architectures and Validation:* Utilizing this expanded design space, we introduce three novel, attention-free, and parallelizable architectures: MONETA, YAAD, and MEMORA. These models combine expressive MLP-based memories with our novel optimization mechanisms. Empirically, we demonstrate that different MIRAS instantiations trade off complementary strengths, achieving strong scaling laws and superior performance compared to state-of-the-art Transformers and linear RNNs across language modeling, commonsense reasoning, and challenging long-context recall tasks.

## 2 ASSOCIATIVE MEMORY, ATTENTIONAL BIAS, AND RETENTION

Associative memory, a core component of human learning (Terry, 2017), has inspired numerous artificial neural architectures (Hopfield, 1982; Schlag et al., 2021; Behrouz et al., 2025c). Broadly speaking, associative memory is an operator that learns mappings between keys and values. To learn these mappings effectively, the memory requires an objective function that measures the quality of the learned associations and guides the learning process. In all formulations, one can set values as identity elements, resulting in a formulation beyond strict design of key-value memory. In fact, while we discuss MIRAS as a framework based on associative memory, the concept of associative memory is not limited to only the design between key-value memory modules and can include design choices such as continuum memory system (CMS) (Behrouz et al., 2025b) that updates their weights based on the objective of the task at hand (e.g., next-token-prediction in language modeling tasks).

**Definition 2.1** (Associative Memory and Attentional Bias). Given a set of keys $\mathcal{K} \subseteq \mathbb{R}^{d_k}$ and values $\mathcal{V} \subseteq \mathbb{R}^{d_v}$, associative memory is an operator $\mathcal{M} : \mathcal{K} \to \mathcal{V}$. Learning the mapping of associative memory is based on an objective $\mathcal{L}$, called *Attentional Bias*, that determines the type of memory and its tendency to prioritize some events:

$$\mathcal{M}^* = \arg\min_{\mathcal{M}} \quad \mathcal{L}(\mathcal{M}(\mathcal{K}); \mathcal{V}). \tag{1}$$

When memory is parameterized by $W$, we use $\mathcal{M}(W, \boldsymbol{k})$. In this setting, the optimization in equation 1 is performed over $W$. This learning process can be viewed as a meta (in-context) learning task, where the model learns how to store data into its parameters at test time (Munkhdalai et al., 2019; Behrouz et al., 2025b).

### 2.1 THE OPTIMIZATION PERSPECTIVE: LEARNING AND RETAINING

Definition 2.1 translates the design of a sequence model into an optimization problem. In an online setting, where key-value pairs $(\boldsymbol{k}_t, \mathbf{v}_t)$ arrive sequentially, a straightforward approach to optimize Equation 1 is to use gradient descent. Given a new pair, we update the memory parameters:

$$W_t = W_{t-1} - \eta_t \nabla \ell(W_{t-1}; \boldsymbol{k}_t, \mathbf{v}_t), \tag{2}$$

where $\ell(W_{t-1}; \boldsymbol{k}_t, \mathbf{v}_t) := \mathcal{L}(\mathcal{M}(W; \boldsymbol{k}_t), \mathbf{v}_t)$. This update can be interpreted as adjusting the memory based on a "momentary surprise" (Behrouz et al., 2025c), where the model prioritizes memorizing tokens that violate the expectations of the objective $\mathcal{L}$. This update rule highlights **a fundamental tension in sequence modeling**: the need to learn from the latest information (adaptability) while remaining stable enough to retain previously memorized context (stability). We can formalize this tension by viewing the gradient descent update (2) through an optimization lens. Mathematically, Equation 2 is equivalent to the solution of the following optimization problem:

$$W_t = \arg\min_{W} \quad \langle W - W_{t-1}, \nabla \ell(W_{t-1}; \boldsymbol{k}_t, \mathbf{v}_t) \rangle + \frac{1}{2\eta_t} \|W - W_{t-1}\|_2^2 \tag{3}$$

The first term locally approximates $\ell(W; \boldsymbol{k}_t, \mathbf{v}_t)$ at the previous state $W_{t-1}$; minimizing it corresponds to learning the new token. The second term is a quadratic penalty that regularizes deviations from $W_{t-1}$; minimizing it corresponds to retaining past information and ensuring stability.

## 2.2 THE LEARNING-RETAINING VIEWPOINT

The formulation in equation 3 relies specifically on linear approximations and quadratic regularization. However, we can generalize this concept by employing different approximations for the attentional bias and alternative functions for retention. This generalization leads to:

$$W_t = \arg\min_{W \in \mathcal{W}} \underbrace{\widetilde{\ell}_t(W; \boldsymbol{k}_t, \mathbf{v}_t)}_{\text{Attentional Bias}} + \underbrace{\text{Ret}_t(W, W_{t-1})}_{\text{Retention}}. \qquad \text{(Learning-Retaining Viewpoint)}$$

Here, $\widetilde{\ell}_t(W; \boldsymbol{k}_t, \mathbf{v}_t)$ is an approximation of $\ell(W; \boldsymbol{k}_t, \mathbf{v}_t)$, driving the learning of new concepts. $\text{Ret}_t(W, W_{t-1})$ is the retention function, regularizing changes in $W$ to maintain stability and preserve learned knowledge. This viewpoint has also been acknowledge by Liu et al. (2024a).

The retention function can be further decomposed into local and global components:

$$\text{Ret}_t(W, W_{t-1}) = \underbrace{\frac{1}{\eta_t} \text{D}_t(W, W_{t-1})}_{\text{Local Retention}} + \underbrace{\frac{1}{\alpha_t} \text{G}_t(W)}_{\text{Global Retention}}.$$

The local retention $\text{D}_t(W, W_{t-1})$ is typically a *premetric* (e.g., $\ell_2$ distance, KL divergence) that controls deviations from the immediate past state $W_{t-1}$. The coefficient $\eta_t$ acts as a meta in-context learning rate, balancing learning (larger $\eta_t$) against retention (smaller $\eta_t$). The global retention $\text{G}_t$ controls the overall complexity or size of the memory (e.g., weight decay).

**Remark on "Forgetting" as Regularization.** Within this viewpoint, mechanisms often termed "forget gates" (Behrouz et al., 2025c; Yang et al., 2024a) are reinterpreted not as explicit erasure mechanisms, but as specific implementations of the Retention function (regularization). The model optimizes how much of the past state to retain by balancing the regularization penalty against the learning objective. **This interpretation is crucial as it provides a principled way to design novel retention mechanisms derived from optimization theory (See Section 4), rather than relying on heuristic gating structures.** Therefore, we use the term *Retention Gate* throughout this work. This interpretation aligns closely with human memory processes, where memories often become inaccessible due to retrieval failures rather than complete erasure (Robertson, 2002).

## 2.3 ALTERNATIVE PERSPECTIVE: FOLLOW-THE-REGULARIZED-LEADER (FTRL)

While the Learning-Retaining viewpoint focuses on the trade-off at the current timestep, an alternative perspective from online optimization considers the entire history of the sequence. The update rule in Equation 2 can also be viewed as one step of online gradient descent on the sequence of losses $\ell(W; \boldsymbol{k}_1, \mathbf{v}_1), \ell(W; \boldsymbol{k}_2, \mathbf{v}_2), \dots, \ell(W; \boldsymbol{k}_t, \mathbf{v}_t), \dots$. Online gradient descent is a special case of the Follow-The-Regularized-Leader (FTRL) algorithm (Shalev-Shwartz et al., 2012; Hazan et al., 2016). In FTRL, the goal is to minimize the cumulative loss over all past tokens, balanced by a regularization term that penalizes the overall complexity of the memory. This leads to:

$$W_t = \arg\min_{W \in \mathcal{W}} \underbrace{\sum_{i=1}^{t} \widehat{\ell}_i(W; \boldsymbol{k}_i, \mathbf{v}_i)}_{\text{Attentional Bias}} + \underbrace{\frac{1}{\eta_t} \mathcal{R}_t(W)}_{\text{Memory Stability}}. \qquad \text{(FTRL Viewpoint)}$$

Here, $\widehat{\ell}_i(W; x_i)$ represents an approximation (e.g., linearization) of the loss at time $i$, and $\mathcal{R}_t(W)$ is the regularization term. The (Learning-Retaining Viewpoint) and (FTRL Viewpoint) offer complementary perspectives (one local, one global) and can formally be connected:

**Theorem 2.2.** *Let $\eta_t = \eta$ and define $h_t(W) := \sum_{i=1}^{t-1} \widehat{\ell}_i(W; \boldsymbol{k}_i, \mathbf{v}_i) + \frac{1}{\eta} R(W)$. Assume $\mathcal{W} = \mathbb{R}^d$ and the function $h_t(W)$ is strictly convex in $W$ and let $\mathcal{D}_h(\cdot, \cdot)$ be the Bregman divergence defined by function $h(\cdot)$, i.e., $\mathcal{D}_h(W, W') = h(W) - h(W') - \langle \nabla h(W'), W - W' \rangle$. Set $\text{Ret}_t(W, W') = \mathcal{D}_h(W, W')$ and $\widetilde{\ell}_t(W; x_t) = \widehat{\ell}_t(W; x_t)$ in (Learning-Retaining Viewpoint). Then, the update rule in (Learning-Retaining Viewpoint) is equivalent to the update rule in (FTRL Viewpoint).*

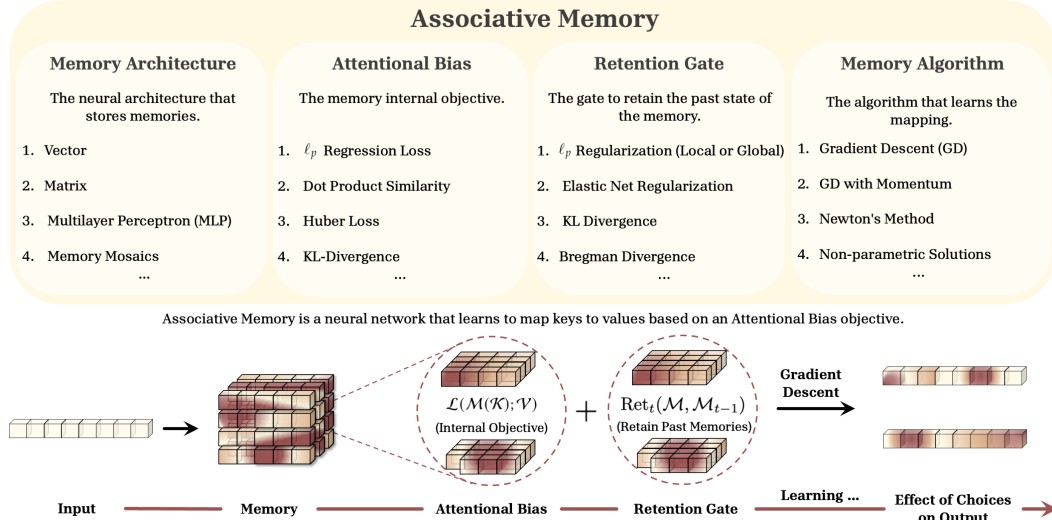

Figure 1: The overview of MIRAS framework. MIRAS is based on four critical choices of (1) memory architecture, (2) attentional bias, (3) retention gate, and (4) memory learning algorithm. In this framework, the memory architecture determines the model capacity to memorize; attentional bias is responsible for modeling the underlying mapping patterns; retention gate determines how to balance learning new concepts and the retention of previously learned concepts; and memory learning algorithm is responsible for memory management.

The proof is provided in Appendix B. Intuitively, Theorem 2.2 confirms that optimizing the cumulative loss over the entire history (FTRL), can be achieved via optimizing for the immediate trade-off between learning and retention in (Learning-Retaining Viewpoint), provided the retention function is appropriately chosen. This suggests that the (Learning-Retaining Viewpoint) is more general. We therefore adopt it as the primary lens for most of our derivations in subsequent sections.

## 2.4 EXTENSIONS AND GENERALIZATIONS

The (Learning-Retaining Viewpoint) can be naturally extended to provide more flexibility. We can consider a *Universal Viewpoint* where the memory update relies on recent history, rather than just the immediate past state:

$$W_t = \arg \min_{W \in \mathcal{W}} \underbrace{\widetilde{\ell}_t(W; \{\boldsymbol{k}_i, \mathbf{v}_i\}_{i=t-k}^t)}_{\text{Attentional Bias}} + \underbrace{\text{Ret}_t\left(W, \{W_{i-1}\}_{i=t-k'}^t\right)}_{\text{Retention}}. \qquad \text{(Universal Viewpoint)}$$

Here, the memory update relies on its recent $k+1$ states and the latest $k'+1$ key-value pairs. While (Learning-Retaining Viewpoint) is a special case of (Universal Viewpoint) by setting $(k, k') = (0, 0)$, one can obtain more flexible designs by choosing higher values of $k, k'$. For example, setting $(k, k') = (0, 1)$ can recover optimization algorithms with momentum, such as those used in Titans (Behrouz et al., 2025c), as discussed in Appendix C.

In all above viewpoints, instead of using the global minimizer of the update, one can approximately find the minimizer by simply using one (or multiple) steps of a particular optimizer. For example, one can consider more advanced optimizers (such as Adam or Muon) for updating memory (Clark et al., 2022; Zhang et al., 2025; Behrouz et al., 2025a). In the next section, we summarize the design choices within the MIRAS framework.

## 3 MIRAS: A FLEXIBLE FRAMEWORK FOR LEARNING TO MEMORIZE

Building viewpoints presented in the previous section, we present MIRAS framework that not only *accurately* unifies existing architectures but also provides insights on how to design the next generation of neural architectures. As discussed in Section 2, learning an associative memory can be interpreted as a meta-learning task, in which the associative memory learns how to compress and store data into its parameters at test time Sun et al. (2024). The architecture of the memory in such task is

particularly important as in longer contexts as the expressivity of the memory structure can limit its ability to learn the underlying patterns. Therefore, the first choice to design a neural architecture is the structure of the memory. Given the structure of the memory, parameterized by a set of parameters $W$, we aim to minimize a loss function $\ell(W; \cdot, \cdot)$ with a retention regularizer $\text{Ret}(\cdot)$ via a learning algorithm (e.g., gradient descent). Accordingly, MIRAS requires four design choices:

1. **Memory Structure:** This choice specifies the architecture of the memory. For example, this architecture can be a linear function, a Multilayer Perceptron (MLP) layer, or even more complex structures. We may restrict the choice of $W$ to be within a certain region, e.g., $W$ to lie within an $L_2$ ball to avoid infinite values or unstable learning.

2. **Attentional Bias:** A key design choice is the objective $\mathcal{L}(\cdot)$ (and consequently its approximations in different viewpoints). This choice determines how we memorize the context and prioritize the events.

3. **Memory Stability and Retention:** Another key choice is the retention regularizer. This choice balances learning with retention of past state. An effective retention gate is key to reliable performance in long context.

4. **Memory Algorithm:** Finally, this choice specifies the learning algorithm that we use to optimize the memory objective. One may use exact minimizer in our framework or use one step (or multiple steps of) a particular optimizer to update the memory.

The design choices of MIRAS are summarized in Figure 1. In Appendix D, we detail how various existing architectures—including Softmax Attention, RNNs with Hebbian rules (e.g., RetNet, Mamba2), RNNs with Delta rules (e.g., DeltaNet), and deep memory models (e.g., Titans)—can be derived as specific instantiations of the MIRAS framework. In particular, Table 1 in the appendix provides a comprehensive overview of this unification. Crucially, this analysis reveals a significant limitation: almost all these methods rely narrowly on $\ell_2$ or dot-product attentional biases and $\ell_2$ retention gates. MIRAS allows other choices of attentional bias/retention gates. In the next section, we discuss how going beyond the standard choices can lead to new architecture designs.

## 4    BEYOND EXISTING ATTENTIONAL BIASES AND RETENTION GATES

As discussed in the previous section, existing work focuses only on linear/quadratic choices for the attentional bias or retention gate. However, in general there could be various choices for all the three aforementioned design choices (even by going beyond Euclidean space). To illustrate the flexibility of our designed framework, this section proposes and discusses novel design choices in MIRAS.

We start by discussing novel choices of attentional biases and retention gates in Section 4.1. We only briefly present three of such variants in this subsection and we will relegate further choices to Appendix E. Then, we combine our design choices in Section 4.2 to obtain three particular architectures: Moneta, Yaad, and Memora, which we further experiment on in Section 5.

### 4.1    NOVEL ALTERNATIVE ATTENTIONAL BIAS AND RETENTION GATES

**Variant 1: $\ell_p$-Attentional Bias.** Attentional bias defines the "similarity metric" and measures how well memory can recall the value, given its corresponding key. Although $\ell_2$ loss is used in prior work, a natural extension is $\ell_p$-norm class of objectives: We define $\ell_p$-attentional bias as:

$$\mathcal{L}(\mathcal{M}(W, \boldsymbol{k}_t); \mathbf{v}_t) = \|\mathcal{M}(W, \boldsymbol{k}_t) - \mathbf{v}_t\|_p^p, \tag{4}$$

where $p \in \mathbb{R}^{\geq 1}$ and $\|.\|_p$ is the $p$-norm. Depending on the distribution of the data, we might want to use different values of $p$ (see Section 5). For the sake of simplicity, let memory be a matrix defining a linear mapping, i.e., $\mathcal{M}(W, \boldsymbol{k}_t) = W\boldsymbol{k}_t$, the gradient descent update is:

$$W_t = W_{t-1} - \eta_t \nabla\ell(W_{t-1}; \boldsymbol{k}_t, \mathbf{v}_t) = W_{t-1} - p\,\eta_t\,\left(\text{Sign}(W_{t-1}\boldsymbol{k}_t - \mathbf{v}_t) \odot |W_{t-1}\boldsymbol{k}_t - \mathbf{v}_t|^{p-1}\right)\,\boldsymbol{k}_t^\top, \tag{5}$$

where $\odot$ is the Hadamard (element-wise) product. For $p = 1$, the recurrence simplifies to: $W_t = W_{t-1} - \eta_t\,\text{Sign}(W_{t-1}\boldsymbol{k}_t - \mathbf{v}_t)\,\boldsymbol{k}_t^\top$. We call this variation *value-less* associative memory, in which we store entities (keys) but map them into two extreme class of -1 and +1 through the sign function. This behavior provides inherent robustness, as the magnitude of the error does not affect the update

direction, preventing extreme events (outliers) from overly influencing the memory. One simple interpretation for such behavior is the coping mechanism in human (Loftus, 1993), in which the memory does not store the values for extreme events. This interpretation of protective memory in extreme events motivates our next variant.

**Variant 2: Huber Loss: Memory with Coping Mechanism.** While $\ell_2$-norm regression objective is a common choice, it is known to be sensitive to noise and extreme examples (outliers). To have a robust loss against outliers, we can use Huber loss as the attention bias, in which an extreme mismatch (potentially due to outlier data) does not affect the memory learning process:

$$W_t = W_{t-1} - \begin{cases} \eta_t \, \nabla\ell_2(W_{t-1}; \boldsymbol{k}_t, \mathbf{v}_t) & \text{if} \quad \|\mathcal{M}(W; \boldsymbol{k}_t) - \mathbf{v}_t\| \leq \delta_t, \\ \eta_t \, \delta_t \nabla\ell_1(W_{t-1}; \boldsymbol{k}_t, \mathbf{v}_t) - \delta_t^2 & \text{Otherwise.} \end{cases} \tag{6}$$

In this formulation, the parameter $\delta_t$ decides the type of the memory ($\ell_2$-norm objective or value-less) based on the context, making the memory more robust to outliers.

**Variant 3: Memorization Over A Scaled Probability Simplex.** To avoid numerical instabilities, we can constrained the variable $W_t$ to lie within a scaled probability simplex. In other words, we can restrict the state to lie in the constraint set $\mathcal{W} = \{W \mid \|W\|_1 = c \text{ and } W_{jl} \geq 0, \ \forall j, l\}$. In this set, each point $W$ can be viewed as a measure. Thus, we can utilize divergences over measures to define our retention in (Learning-Retaining Viewpoint) . For example, by choosing $\mathrm{D}_t(W, W')$ as the KL divergence and setting $\widetilde{\ell}(W; \boldsymbol{k}_t, \mathbf{v}_t) = \langle W - W_{t-1}, \nabla\ell(W_{t-1}; \boldsymbol{k}_t, \mathbf{v}_t)\rangle$ in (Learning-Retaining Viewpoint) , we get the update rule

$$W_t \leftarrow c \, \text{Softmax} \left((1 - \lambda) \log(W_{t-1}) - \eta\nabla\ell(W_{t-1}; \boldsymbol{k}_t, \mathbf{v}_t)\right) \tag{7}$$

where $\lambda \in (0, 1)$ and $\eta \in \mathbb{R}^+$ are the hyper-parameters that can be learned during training. The Softmax operator ensures that the output lies in the set $\mathcal{W}$. To see more discussions and extensions to general f-divergence retention gates, see Section E.2 in the appendix.

**Other Variants.** In the appendix section, we derive other novel variants of MIRAS by using elastic net regularization, Bregman divergence and $f-$ divergence retention gates, and $L_q$ memory stability. We will also discuss the above variants in more details.

### 4.2 Focus Variants of Miras: Moneta, Yaad, and Memora

Using the above basic variants and the other variants explained in Appendix E, we now introduce three instantiations of MIRAS, each designed to explore different facets of this expanded optimization space, moving beyond the standard $\ell_2$ and dot-product paradigms.

**MONETA.** MONETA is designed to investigate the impact of generalized norms. Given $p, q \in \mathbb{R}^{\geq 1}$, we design $(p, q)$-MONETA to explore the impact of generalized $\ell_p$ norms for both learning and regularization. We instantiate MIRAS as follows. *Memory Structure:* A 2-layer MLP with expansion factor 4, GELU activation (Hendrycks & Gimpel, 2016), residual connections, and Layer Norm $\mathcal{M}(x) = x + \text{LN}(W_1\sigma(W_2x))$. *Attentional Bias:* $\ell_p$ norm. *Retention Gate:* A hybrid of $\ell_q$ retention gate $\frac{1}{2(q-1)}\|W\|_q^2$ (see Appendix E for details) and the standard $\ell_2$ regularization $\frac{1}{\alpha_t}\|W\|_2^2$. *Memory Algorithm:* Gradient Descent. The above choices result in the following recurrent formula for the memory module:

$$A_t = \beta_t A_{t-1} - \eta_t \nabla\ell(W_{t-1}; \boldsymbol{k}_t, \mathbf{v}_t), \quad \text{and} \quad W_t = \frac{A_t}{\|A_t\|_q^{q-2}}. \tag{8}$$

Notably the gradient can be calculated using Equation 5. We use $(p, q) = (3, 4)$.

**YAAD.** YAAD is designed for robustness, protecting the memory from extreme events (outliers) using principles from robust statistics. We design YAAD based on the Huber objective. *Memory Structure:* MLP (same architecture as MONETA). *Attentional Bias:* Huber loss (Equation 6). *Retention Gate:* A combination of local and global retention: $\text{Ret}_t(W, W_{t-1}) = \frac{1}{2\eta_t}\|W - W_{t-1}\|_F^2 + \frac{1}{\alpha_t}\|W\|_2^2$. *Memory Algorithm:* Gradient Descent. Given these choices, we can write the resulting memory learning process as :

$$W_t = \beta_t W_{t-1} - \begin{cases} \eta_t \, \nabla\ell_2(W_{t-1}; \boldsymbol{k}_t, \mathbf{v}_t) & \text{if} \quad \|\mathcal{M}(\boldsymbol{k}_t) - \mathbf{v}_t\| \leq \delta_t, \\ \eta_t \, \delta_t \nabla\ell_1(W_{t-1}; \boldsymbol{k}_t, \mathbf{v}_t) - \delta_t^2 & \text{Otherwise.} \end{cases} \tag{9}$$

Table 2: Performance of MIRAS's variants and baselines on language modeling and common-sense reasoning tasks. Hybrid models are marked with *. The best results are highlighted.

| Model | Wiki. ppl↓ | LMB. ppl↓ | LMB. acc↑ | PIQA acc↑ | Hella. acc_n↑ | Wino. acc↑ | ARC-e acc↑ | ARC-c acc_n↑ | SIQA acc↑ | BoolQ acc↑ | Avg. ↑ |
|---|---|---|---|---|---|---|---|---|---|---|---|
| 1.3B params / 100B tokens | | | | | | | | | | | |
| Transformer++ | 18.53 | 18.32 | 42.60 | 70.02 | 50.23 | 53.51 | 68.83 | 35.10 | 40.66 | 57.09 | 52.25 |
| RetNet | 19.08 | 17.27 | 40.52 | 70.07 | 49.16 | 54.14 | 67.34 | 33.78 | 40.78 | 60.39 | 52.02 |
| Mamba2 | 16.56 | 12.56 | 45.66 | 71.87 | 55.67 | 55.24 | 72.47 | 37.88 | 40.20 | 60.13 | 54.89 |
| DeltaNet | 17.71 | 16.88 | 42.46 | 70.72 | 50.93 | 53.35 | 68.47 | 35.66 | 40.22 | 55.29 | 52.14 |
| Gated DeltaNet | 16.42 | 12.17 | 46.65 | 72.25 | 55.76 | 57.45 | 71.21 | 38.39 | 40.63 | 60.24 | 55.32 |
| Samba* | 16.13 | 13.29 | 44.94 | 70.94 | 53.42 | 55.56 | 68.81 | 36.17 | 39.96 | 62.11 | 54.00 |
| Gated DeltaNet-H2* | 15.91 | 12.55 | 48.76 | 72.19 | 56.88 | 57.77 | 71.33 | 39.07 | 41.91 | 61.55 | 56.18 |
| Titans (LMM) | 15.60 | 11.41 | 49.14 | 73.09 | 56.31 | 59.81 | 72.43 | 40.82 | 42.05 | 60.97 | 56.82 |
| MONETA (ours) | 15.52 | 11.47 | 47.88 | 73.16 | 56.14 | 59.09 | 72.53 | 40.32 | 41.91 | 61.18 | 56.52 |
| YAAD (ours) | 15.18 | 11.89 | 47.23 | 72.81 | 56.46 | 59.02 | 72.14 | 40.05 | 40.73 | 61.86 | 56.39 |
| MEMORA (ours) | 15.90 | 12.04 | 48.67 | 73.10 | 55.99 | 57.36 | 71.55 | 37.92 | 40.19 | 61.34 | 55.87 |

Note that for improving the expressive power, in all architectures, we decouple the learning rate $\eta$ and the retention gate rate $\alpha$, resulting in a independent parameter $\beta_t \in [0, 1]$.

**MEMORA.** Finally, MEMORA is designed to ensure stable updates by constraining the memory to a probability simplex and utilizing divergence-based retention. *Memory Structure:* MLP (same as MONETA), constrained to the scaled probability simplex. *Attentional Bias:* dot-product loss. *Retention Gate:* KL-divergence for local retention and Shannon entropy for global retention (Appendix E). *Memory Algorithm:* Closed-form solution. These choices lead to (see equation 30):

$$W_t = \text{Softmax}\left(\beta_t \log(W_{t-1}) - \eta_t \nabla\ell(W_{t-1}; \boldsymbol{k}_t, \mathbf{v}_t)\right) \tag{10}$$

**Architecture Backbone.** For the architectural backbone, we fully follow recent studies (Behrouz et al., 2025c; Yang et al., 2024a): We replace attention modules with our variants of MIRAS in Llama's macro architecture with MLPs with `SwiGLU`(.) activation, rotary positional encodings (RoPE) (Su et al., 2024), and RMSNorm (Zhang & Sennrich, 2019). We incorporate a 1D depthwise-separable convolution layer after each of the query, key, and value projections. For training stability, we also use $\ell_2$ normalization to $q$ and $k$. The output of this module is normalized and gated with a linear layer (Mehta et al., 2023). For all input-dependent parameters like $\eta_t, \beta_t$, and $\delta_t$, we define them as the linear projection of the input. The architectures are illustrated in Figure 5.

**Parallelizable Training.** We build upon the work of Behrouz et al. (2025c); Sun et al. (2024) and use a hybrid recurrence of linear and non-linear by chunking the sequences into small subsequences. While the use of MLP memories and non-Euclidean optimization introduces non-linearities in the recurrence, the hybrid chunking strategy (Appendix F) ensures that the training remains highly parallelizable. Inside each chunk, the recurrence is effectively linearized, and non-linear operations (e.g., the normalization in MONETA or Softmax in MEMORA) are applied only at chunk boundaries. This maintains competitive training throughput while offering $O(1)$ complexity per token during inference.

## 5 EXPERIMENTS

Experimental details (resp. additional experiments) are in Appendix G (resp. Appendix H).

### 5.1 LANGUAGE MODELING AND COMMON-SENSE REASONING

We follow recent studies (Yang et al., 2024a;c; Behrouz et al., 2025c) and first focus on the perplexity in language modeling and commonsense reasoning tasks. The results for MEMORA, YAAD, MONETA and baselines with size of 1.3B are reported in Table 2 (Full results of 340M and 760 in Table 7). All of our variants outperforms all the baselines including Transformer++, modern linear recurrent models and hybrid methods. The superior performance compared to hybrid models is particularly important as all our variants are pure recurrent (attention-free). Among the three variants of MIRAS, while MONETA achieves slightly weaker performance than MEMORA, and YAAD, the other two variants are close and depending on the task and model size, the best model can vary.

### 5.2 SCALING PATTERN

To evaluate the scaling pattern of models and for comparing them with baseline, in this section, we plot their performance with varying the model size and the context window.

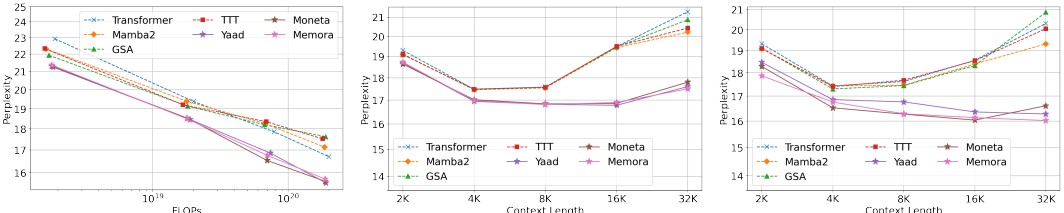

Figure 2: Scaling patterns when increasing (**Left**) model size, (**Middle**) sequence length (model size = 340M) (3) (**Right**) sequence length (model size = 760M) on C4 dataset.

**Context Length.** We first vary the training context length from 2K to 32K for two version of our model with size 340M and 760M. The results are reported in Figure 2 (Middle and Right). All three variants of MIRAS scales better than state-of-the-art baselines when increasing the context length. We attribute this superior performance to: (1) expressive memory architecture. Contrary to baselines like Mamba2 and GSA that uses vector- and matrix-valued memory, our variants use 2-layer MLPs with more expressive power. (2) The choice of retention gate and attentional bias. While TTT also uses MLP memory, it shows weaker scaling. **This highlights a crucial finding: expressive memory alone is insufficient; it requires correct design choices (e.g. attentional bias, retention, and optimization algorithm) to be effectively utilized.** All of our three variants go beyond the standard $\ell_2$-based attentional biases and retention gates. These robust choices prevent memory corruption from outliers or noise, leading to better utilization of the fixed capacity, especially in long contexts.

**Model Size.** We also report the #FLOPs vs. perplexity of our models and baselines in Figure 2 (Left). All three variants outperforms all baselines given almost the same budget of FLOPs. These results, once again support the importance of powerful memory design.

## 5.3 NEEDLE IN HAYSTACK

To evaluate the effective context window of our models and baselines, we use needle-in-a-haystack task. In this task, we evaluate the model on retrieving a piece of information (i.e., the "needle") from long distractor texts (i.e., the "haystack"). We focus on the Single NIAH (S-NIAH) task from RULER benchmark (Hsieh et al., 2024) and evaluate our models and baselines on sequences with length 1K, 2K, 4K, and 8K. The results are reported in Table 3. All our variants outperforms the baselines by a considerable margin. Interestingly, MONETA shows superior performance when the data is synthetic noise (S-NIAH-PK). This highlights the advantage of MONETA's $\ell_p$-attentional bias and $\ell_q$ retention (with $(p, q) = (3, 4)$). **Unlike the $\ell_2$ objectives used in baselines, these higher-order norms are inherently more robust to noisy inputs, preventing the distractor texts from corrupting the memory state.** This validates the effectiveness of exploring non-Euclidean design choices via MIRAS.

Table 3: Performance of MONETA, YAAD, MEMORA, and baselines on NIAH task from RULER benchmark. The best results with highest accuracy are highlighted.

Table 4: Ablation on the architecture of MEMORA and MONETA.

| Model | S-NIAH-PK | | | S-NIAH-N | | | S-NIAH-W | | | Average |
|---|---|---|---|---|---|---|---|---|---|---|
| | 2K | 4K | 8K | 2K | 4K | 8K | 1K | 2K | 4K | |
| Mamba2 | 98.6 | 61.4 | 31.0 | 98.4 | 55.8 | 14.2 | 62.2 | 42.2 | 4.2 | 52.0 |
| DeltaNet | 96.8 | 98.8 | 98.6 | 47.2 | 15.4 | 12.8 | 85.2 | 46.2 | 20.0 | 57.9 |
| Gated DeltaNet | 89.8 | 91.4 | 90.0 | 99.2 | 91.8 | 26.4 | 86.4 | 82.6 | 24.4 | 75.8 |
| TTT | 98.4 | 98.8 | 98.0 | 60.2 | 36.6 | 10.2 | 85.8 | 78.8 | 28.0 | 66.1 |
| MONETA | 99.4 | 98.8 | 98.8 | 99.4 | 99.4 | 92.8 | 92.2 | 88.2 | 70.8 | 93.5 |
| YAAD | 99.2 | 98.6 | 94.4 | 99.8 | 98.6 | 93.2 | 91.8 | 89.6 | 67.4 | 92.9 |
| MEMORA | 99.2 | 98.8 | 92.6 | 98.4 | 99.2 | 93.2 | 92.4 | 88.2 | 70.4 | 92.1 |

| Variant | MEMORA | MONETA |
|---|---|---|
| Full Architecture | 51.52 | 52.12 |
| w/o Retention Gate | 49.75 | 50.49 |
| linear memory | 50.11 | 50.26 |
| w/o RoPE | 51.28 | 51.71 |

## 5.4 ABLATION STUDY

We perform ablation studies on 760M model to validate if different design choices we discussed through the paper are positively contributing to better performance. Additional ablations are in Appendix H.

**The Effect of Design.** To evaluate the architectural design choices, we perform an ablation study on MEMORA, and MONETA in Table 4, as well as on YAADin Table 5. The first row, reports the performance of full architecture, while (1) the second row removes the retention (i.e., $\beta = 1$), and (2) third row replaces the MLP with a linear layer. In Table 5, (3) forth row makes $\delta$ input independent, (4) the next row removes $\ell_2$-loss from the Huber loss, and (5) the last row removes the $\ell_1$ condition. These results indicate that all design choices are contributing to the performance of the model.

**The Effect of $p$ on Performance.** We first evaluate the effect of $p$ on the performance of MONETA. We vary the value of $p \in \{1, 1.5, 2, 2.8, 3, 3.2, 4\}$ and context window from 2K to 16K. The results are reported in Figure 3. Interestingly, there is no monotone pattern when increasing the value of $p$ and the best performance is achieved when $p = 3$, while $p = 4$ achieves the worst performance. Also, although different values of $p$ results in different memory modules with varied performance, the scaling pattern when increasing the context length is almost the same.

**The Effect of $q$ on Performance.** We evaluate the effect of $q$ by varying its value in $\{2, 3, 4, 5\}$. Interestingly, contrary to $p$, the value of $q$ can change the scaling pattern when increasing the context length. The main reason for this observation is that the value of $q$ determines the retention gate and a powerful retention gate can improve the memory management, resulting in better performance in longer context.

Table 5: Ablation study on the components of YAAD.

| Model | Avg. LM |
|---|---|
| YAAD | 53.98 |
| - Retention Gate | 50.63 |
| linear memory | 51.57 |
| - Input-dependent $\delta$ | 52.19 |
| $\ell_2$-loss | 52.86 |
| $\ell_1$-loss | 53.04 |

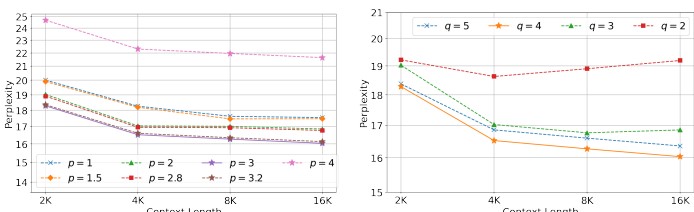

Figure 3: The effect of parameters $p$ and $q$ on the performance with different context length.

## 6 LIMITATIONS AND DISCUSSIONS

While MIRAS framework provides a principled approach to designing efficient sequence models with strong performance in language modeling and long-context recall (e.g., Needle-in-a-Haystack), our variants still lag behind quadratic Transformer models in complex in-context retrieval tasks (e.g., Table 9). This highlights a known challenge for recurrent models, suggesting that while optimized memory management can replace attention in many scenarios, perfect in-context recall remains an area for future improvement, mainly due to the fact that it is more a challenge for the fixed-size memory of RNNs rather than its memory update mechanism. One might combine the design choices in MIRAS with techniques like Memory Caching (Behrouz et al., 2026) to achieve recurrent models with growing memory and effective memory management. Furthermore, while the framework connects learning in non-convex memories (MLPs) to online optimization, providing theoretical guarantees on convergence and learning in this setting remains challenging.

## 7 CONCLUSION

This paper presents MIRAS, a general framework that explains the connection of online optimization and test time memorization. MIRAS framework can explain the role of several standard architectural choices in the literature (e.g., forget gate) and helps design next generation of architectures that are capable of managing the memory better. Building upon our framework, by using novel attentional bias and retention gates, we present three novel sequence models–called YAAD, MONETA, and MEMORA–each of which with its own (dis)advantages. Our experimental evaluations show that all these variants outperform various baselines in various downstream tasks.

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

## A    PRELIMINARIES AND BACKGROUND

In this section, we review the related studies and background concepts that we use through the paper.

**Attention.** Attention as the backbone of Transformers is a critical component that acts as their associative memory (Bietti et al., 2023). Given input $x \in \mathbb{R}^{N \times d_{\text{in}}}$, causal attention computes output $\mathbf{y} \in \mathbb{R}^{N \times d_{\text{in}}}$ based on Softmax over input dependent key, value, and query matrices:

$$\mathbf{Q} = x\mathbf{W_Q}, \qquad \mathbf{K} = x\mathbf{W_K}, \qquad \mathbf{V} = x\mathbf{W_V}, \tag{11}$$

$$\mathbf{y}_i = \sum_{j=1}^{i} \frac{\exp\left(\mathbf{q}_i^\top \mathbf{k}_j / \sqrt{d_{\text{in}}}\right) \mathbf{v}_j}{\sum_{\ell=1}^{i} \exp\left(\mathbf{q}_i^\top \mathbf{k}_\ell / \sqrt{d_{\text{in}}}\right)}, \tag{12}$$

where $\mathbf{W_Q}, \mathbf{W_K}$, and $\mathbf{W_V} \in \mathbb{R}^{d_{\text{in}} \times d_{\text{in}}}$ are learnable parameters. While Transformers achieve significant improvements compared to traditional Recurrent Neural Networks (RNNs)—such as LSTM (Schmidhuber & Hochreiter, 1997), their complexity that requires at least $N \times d$ operators to calculate the output has been the main motivation for researchers to think about alternative architectures. We divide and review the research efforts to design alternative architectures into two groups: (1) Linear shallow memory recurrent models, (2) Deep memory modules.

**(Linear) Recurrent Models.** For many years, non-linear (gated) recurrent neural networks had been the de facto architectural backbones in deep learning (Greff et al., 2016). Their recurrent nature, however, results in non-parallelizable training, making their large scale training infeasible. To this end, in recent years, linear RNNs as alternatives to both Transformers and non-linear RNNs attract much attention mainly due to their parallelizable and linear-time training while maintaining competitive performance (Yang et al., 2024c; Sun et al., 2023; Peng et al., 2025a). Earlier variants of linear RNNs (Yang et al., 2024b; Sun et al., 2023; De et al., 2024), which mostly are based on Hebbian learning rule (Hebb, 2005), aim to compress the data into their vector-valued (or matrix-valued) memory (Katharopoulos et al., 2020; Sun et al., 2023; Yang et al., 2024b; De et al., 2024; Liu et al., 2024a). Let $\mathcal{M}_t \in \mathbb{R}^{d \times n}$ be the memory ($n = 1$ means vector-valued memory), and $\mathbf{k}, \mathbf{v} \in \mathbb{R}^d$ are keys and values (i.e., projection of input $x_t \in \mathbb{R}^d$), a simple general formulation for such linear RNNs can be written as:

$$\mathcal{M}_t = A_t * \mathcal{M}_{t-1} + \mathbf{v}_t \mathbf{k}_t^\top, \tag{13}$$

where $*$ is an arbitrary associative operator and $A_t$ is a data-(in)dependent diagonal matrix or a scalar (Yang et al., 2024c). Despite the efficiency that comes with the *linear* recurrent nature of these models, the memory can overflow mainly due to the additive (without replacement) nature of Hebbian learning rule, resulting in limited memory capacity and limited expressive power in in-context learning tasks. Moreover, the vector-valued memory of these architectures can limited their ability to learn/memorize large context window, mainly due to the limited expressive power of memory to learn the underlying patterns of data (Behrouz et al., 2025c; Sun et al., 2024).

To address the above mentioned limitations, recurrent models that use a matrix-valued memory with Delta learning rule has gained popularity in recent years (Schlag et al., 2021; Yang et al., 2024c). Despite significant advantages, even these delta-rule-based recurrent models face theoretical limitations (Irie et al., 2023) with moderate performance in practice (Yang et al., 2024c). Recently, several studies aim to improve the performance of such models by adding scalar or channel-wise forget gate mechanisms (Yang et al., 2024a; Peng et al., 2025b), , using negative eigenvalues (Grazzi et al., 2024), and multiple learning steps (Siems et al., 2025). They, however, still suffer from performance drop in long context, mainly due to the less expressive memory architectures (Behrouz et al., 2025c).

**Deep Memory Module: Titans and Test Time Training.** To overcome the limited memory and to extend the *effective* context length of deep sequence models, more recent studies focus on a new generation of architectures with deep memory module (Behrouz et al., 2025c; Sun et al., 2024). These architectures are built on the meta-learning perspective, where the memory is an MLP architecture that is updated using gradient descent (with momentum) (Behrouz et al., 2025c; Sun et al., 2024). Sun et al. (2024) further provide a unifying perspective that how linear and softmax attention are respectively parametric and non-parameteric solutions of (kernel) regression loss but consider other

modern linear RNNs outside of this class of models. Recently, in a concurrent work to ours, Wang et al. (2025) show that with additional simplification of modern RNNs (e.g., RetNet (Sun et al., 2023), Mamba (Dao & Gu, 2024)) they approximately place in the same class of models that internally optimize regression loss. It, however, still remains unanswered that "What is the underlying design framework behind these sequence models that can *accurately* unify existing architectures?" Moreover, the role of forget gates and its alternative choices in modern sequence models is surprisingly less explored.

To clarify the relationships among existing architectures, several recent works have sought unifying perspectives. Liu et al. (2024) adopt an online-learner view, closely aligned with our (Learning-Retaining Viewpoint) , while the concurrent work of Wang et al. (2025) frames the problem as online regression, which corresponds to our (FTRL Viewpoint) . Our approach formally links these two viewpoints (Theorem 2.2). Unlike those studies, which restrict themselves to $\ell_2$ and dot-product losses, MIRAS extends beyond these standard choices: it supports non-Euclidean losses and regularizations, enabling new architectural designs. This makes MIRAS a comprehensive framework that both (i) explicitly interprets retention/forget gates as forms of regularization and (ii) generalizes the attentional-bias objective beyond simple regression losses.

## B   PROOF OF THEOREM 2.2

Here we present the proof of Theorem 2.2. For the sake of completeness, let us first re-state this Proposition.

**Theorem 2.2.** Let $\eta_t = \eta$ and define $h_t(W) := \sum_{i=1}^{t-1} \widehat{\ell}_i(W; \boldsymbol{k}_i, \boldsymbol{v}_i) + \frac{1}{\eta} R(W)$. Assume $\mathcal{W} = \mathbb{R}^d$ and the function $h_t(W)$ is strictly convex in $W$ and let $\mathcal{D}_h(\cdot, \cdot)$ be the Bregman divergence defined by function $h(\cdot)$, i.e., $\mathcal{D}_h(W, W') = h(W) - h(W') - \langle \nabla h(W'), W - W' \rangle$. Set $\text{Ret}_t(W, W') = \mathcal{D}_h(W, W')$ and $\widetilde{\ell}_t(W; x_t) = \widehat{\ell}_t(W; x_t)$ in (Learning-Retaining Viewpoint) . Then, the update rule in (Learning-Retaining Viewpoint) is equivalent to the update rule in (FTRL Viewpoint) .

*Proof.* Let $\{\widehat{W}_1, \widehat{W}_2, \ldots\}$ be the sequence of parameters obtained by (FTRL Viewpoint) and $\{\widetilde{W}_1, \widetilde{W}_2, \ldots\}$ be the sequence of parameters obtained by (Learning-Retaining Viewpoint) . To show both update rules are equivalent, it suffices to show that the above two sequences are the same if they are initialized at the same point. We prove this statement by induction. First of all, since both sequences are initialized at the same point, the induction base is satisfied (i.e. $\widetilde{W}_1 = \widehat{W}_1$. Now, assume by induction hypothesis that

$$\widetilde{W}_{t-1} = \widehat{W}_{t-1}. \tag{14}$$

To complete the induction, we need to show $\widetilde{W}_t = \widehat{W}_t$. To this end, notice that, by (Learning-Retaining Viewpoint) , we have

$$\widetilde{W}_t = \arg\min_W \quad \widetilde{\ell}_t(W, \boldsymbol{k}_t, \boldsymbol{v}_t) + \text{Ret}_t(W, \widetilde{W}_{t-1})$$

Using the choice of the Attentional Bias and the Retention function in the Proposition, we obtain

$$
\begin{aligned}
\widetilde{W}_t = \arg\min_W \quad & \widehat{\ell}_t(W, \boldsymbol{k}_t, \boldsymbol{v}_t) + \sum_{i=1}^{t-1} \widehat{\ell}_i(W, \boldsymbol{k}_i, \boldsymbol{v}_i) + \frac{1}{\eta} R(W) - \sum_{i=1}^{t-1} \widehat{\ell}_i(\widetilde{W}_{t-1}, \boldsymbol{k}_i, \boldsymbol{v}_i) \\
& - \frac{1}{\eta} R(\widetilde{W}_{t-1}) - \left\langle \sum_{i=1}^{t-1} \nabla \widehat{\ell}_i(\widetilde{W}_{t-1}, \boldsymbol{k}_i, \boldsymbol{v}_i) + \frac{1}{\eta} \nabla R(\widetilde{W}_{t-1}), W - \widetilde{W}_{t-1} \right\rangle.
\end{aligned}
\tag{15}
$$

Ignoring the constant terms and using the induction hypothesis equation 14, we get

$$
\begin{aligned}
\widetilde{W}_t = \arg\min_W \quad & \widehat{\ell}_t(W, \boldsymbol{k}_t, \boldsymbol{v}_t) + \sum_{i=1}^{t-1} \widehat{\ell}_i(W, \boldsymbol{k}_i, \boldsymbol{v}_i) + \frac{1}{\eta} R(W) \\
& - \left\langle \sum_{i=1}^{t-1} \nabla \widehat{\ell}_i(\widehat{W}_{t-1}, \boldsymbol{k}_i, \boldsymbol{v}_i) + \frac{1}{\eta} \nabla R(\widehat{W}_{t-1}), W - \widehat{W}_{t-1} \right\rangle.
\end{aligned}
\tag{16}
$$

On the other hand, recall that $\{\widehat{W}_1, \widehat{W}_2, \ldots\}$ is obtained by (FTRL Viewpoint) . Therefore, we have

$$\widehat{W}_{t-1} = \arg\min_W \quad \sum_{i=1}^{t-1} \widehat{\ell}_i(W; \boldsymbol{k}_i, \boldsymbol{v}_i) + \frac{1}{\eta} \mathcal{R}_t(W).$$

Thus, we have

$$\sum_{i=1}^{t-1} \nabla\widehat{\ell}_i(W_{t-1}, \boldsymbol{k}_i, \boldsymbol{v}_i) + \frac{1}{\eta}\nabla R(W_{t-1}) = 0. \tag{17}$$

Combining equation 17 and equation 16, we obtain

$$\widetilde{W}_t = \arg\min_W \quad \sum_{i=1}^{t} \widehat{\ell}_i(W, \boldsymbol{k}_i, \boldsymbol{v}_i) + \frac{1}{\eta} R(W).$$

This implies $\widetilde{W}_t = \widehat{W}_t$, which completes the proof. $\qquad\square$

## C  VIEWING TITANS AS (UNIVERSAL VIEWPOINT)

Here we discuss how Titans in Behrouz et al. (2025c) can be viewed as a special instantiation of the (Universal Viewpoint). Let $(k, k') = (0, 1)$. Set

$$\widetilde{\ell}_t(W; \{\boldsymbol{k}_i, \mathbf{v}_i\}_{i=t-k}^t) = \langle W - W_{t-1}, \nabla\ell(W_{t-1}, \boldsymbol{k}_t, \mathbf{v}_t)\rangle$$

and

$$\mathrm{Ret}_t\left(W, \{W_{i-1}\}_{i=t-k'}^t\right) = \frac{1}{2\theta_t} \left\| W - \left((1 - \alpha_t + \eta_t)W_{t-1} - \eta_t(1 - \alpha_t)W_{t-2}\right) \right\|^2$$

in (Universal Viewpoint). Then, it is not hard to verify that the update rule for $W_t$ can be given as

$$W_t = (1 - \alpha_t + \eta_t)W_{t-1} - \eta_t(1 - \alpha_t)W_{t-2} - \theta_t\nabla\ell(W_{t-1}, \boldsymbol{k}_t, \mathbf{v}_t).$$

This dynamics is equivalent to

$$W_t = (1 - \alpha_t)W_{t-1} + S_t$$
$$S_t = \eta_t S_{t-1} - \theta_t\nabla\ell(W_{t-1}, \boldsymbol{k}_t, \mathbf{v}_t),$$

which is essentially the gradient descent update with momentum used in Titans of Behrouz et al. (2025c).

## D  UNIFYING VARIOUS EXISTING METHODS UNDER MIRAS FRAMEWORK

In this section, we discuss how various existing architectures fit into MIRAS framework. To facilitate the discussion, we recall Figure 1 for comprehensive presentation of MIRAS.

Next, we discuss how various existing architectures can be unified under MIRAS.

**RNNs with Hebbian Rule.** The first generation of modern recurrent architectures (e.g., Linear attention (Katharopoulos et al., 2020), RetNet (Sun et al., 2023), Mamba (Gu & Dao, 2024), and GLA (Yang et al., 2024b)) are based on Hebbian-like (e.g., gated Hebbian) learning rule (Hebb, 2005). We let attentional bias be the dot product similarity. That is, given a memory $\mathcal{M} \in \mathbb{R}^{d \times n}$ and $\boldsymbol{k}, \mathbf{v} \in \mathbb{R}^d$, we define $\widetilde{\ell}_t := -2\langle \mathcal{M}_t\boldsymbol{k}_t, \mathbf{v}_t\rangle$ and *local retention* as $\mathrm{Ret}_t(\mathcal{M}, \mathcal{M}_{t-1}) = \|\mathcal{M}_t - \alpha\mathcal{M}_{t-1}\|_F^2$. Using Equation Learning-Retaining Viewpoint and gradient descent as the optimizer (i.e., memory learning algorithm), the memory update rule is:

$$\mathcal{M}_t = \alpha\mathcal{M}_{t-1} + \mathbf{v}_t\boldsymbol{k}_t^\top. \tag{18}$$

When (1) $\alpha = 1$, memory update is equivalent to Linear Attention (LA) (Katharopoulos et al., 2020); (2) $\alpha \in \mathbb{R}$ is a learnable parameter, resulting architecture is either lightening attention ($n > 1$) (Li et al., 2025) or RetNet ($n = 1$) (Sun et al., 2023); and (3) $\alpha_t \in \mathbb{R}$ are *data-dependent* learnable parameters, resulting sequence model is Mamba2 (Dao & Gu, 2024).

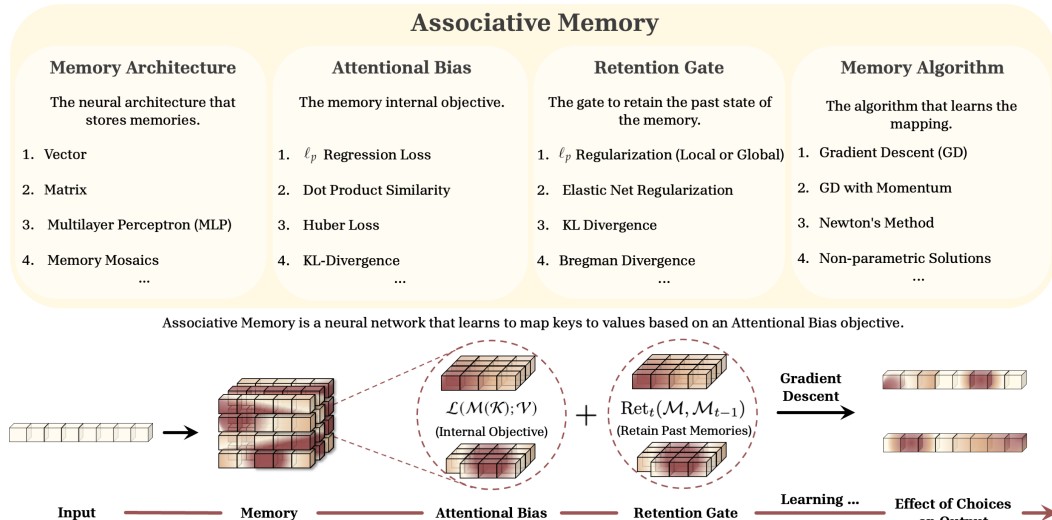

Figure 4: The overview of MIRAS framework. MIRAS is based on four critical choices of (1) memory architecture, (2) attentional bias, (3) retention gate, and (4) memory learning algorithm. In this framework, the memory architecture determines the model capacity to memorize; attentional bias is responsible for modeling the underlying mapping patterns; retention gate determines how to balance learning new concepts and the retention of previously learned concepts; and memory learning algorithm is responsible for memory management.

**RNNs with Delta Rule.** To improve the memory management and to enhance the memory capacity of the above group, several studies suggest using delta rule (Schlag et al., 2021) as the learning algorithm in recurrent neural networks (e.g., DeltaNet (Schlag et al., 2021), Longhorn (Liu et al., 2024a), and RWKV7 (Peng et al., 2025b)). In this part, we recall that where $\mathcal{M} \in \mathbb{R}^{d \times n}$, delta rule is equivalent to optimizing MSE objective $\|\mathcal{M}_t \boldsymbol{k}_t - \mathbf{v}_t\|_2^2$ with $\text{Ret}_t(\mathcal{M}, \mathcal{M}_{t-1}) = \|\mathcal{M}_t - \alpha \mathcal{M}_{t-1}\|_F^2$ as local retention, and stochastic gradient descent as optimizer: ($\eta_t$ is defined in Equation Learning-Retaining Viewpoint)

$$\mathcal{M}_t = \alpha \left( \mathbf{I} - \eta_t \boldsymbol{k}_t \boldsymbol{k}_t^\top \right) \mathcal{M}_{t-1} + \mathbf{v}_t \boldsymbol{k}_t^\top. \tag{19}$$

When (1) $\alpha = 1$, memory update is equivalent to DeltaNet (Schlag et al., 2021); and (2) $\alpha_t \in \mathbb{R}^m$ are *data-dependent* learnable parameters, resulting sequence model is Gated DeltaNet (Yang et al., 2024a) (when $m = 1$). Therefore, RNNs with delta rule are special instances of MIRAS.

**Beyond Delta Rule.** As discussed earlier, while delta rule with its value replacement strategy is more powerful than Hebbian-like learning rules, it suffers from theoretical limitations (Irie et al., 2023) and achieves moderate performance in practice (Yang et al., 2024c). Therefore, several studies have focused on update rules beyond delta rule. Recently, Titans (Behrouz et al., 2025c) suggests using non-linear MSE objective of $\|\mathcal{M}_t(\boldsymbol{k}_t) - \mathbf{v}_t\|_2^2$ with both local and global retention of $\mathrm{D}_t = \|W_t - W_{t-1}\|_F^2$ and $\mathrm{G}_t = \|W_t\|_2^2$ and optimize it with gradient descent with *momentum* [2]. Therefore, Titans-LMM is a special instance of MIRAS, where we use the abovementioned attentional bias and retention regularizations, and gradient descent with momentum as the optimizer. Another way to obtain Titans under MIRAS is explained in Appendix C.

Another example of such models is Mesa-layer (Von Oswald et al., 2023; von Oswald et al., 2025), in which the model uses $\sum_{i=1}^t \|\mathcal{M}_t(\boldsymbol{k}_i) - \mathbf{v}_i\|_2^2$ as the attentional bias objective with $\|\mathcal{M}_t\|_2^2$ as the retention regularization. Since these models uses Newton's method to optimize such an objective, they provide a more expressive update rule than delta rule. We further discuss a set of new learning algorithms beyond delta rule in Section 4.

---

[2]The retention gate (forget gate) in Titans is different from Mamba2 and Gated DeltaNet that we discussed above. The main difference comes from the case of full memory erase. While Mamba2 gating removes the entire memory and treats the next token as the first ever seen data, Titans use a "*cold start*" strategy and use previous state of the memory to measure the surprise of the incoming token before fully erasing the memory.

**Attention.** As discussed by Sun et al. (2024), softmax attention is a non-parameteric solution of $\ell_2$-MSE loss function (i.e., $\|W\boldsymbol{k} - \mathbf{v}\|_2^2$) with Nadaraya-Watson estimator. Therefore, softmax attention (i.e., Transformers) is an instance of MIRAS, when we find the non-parameteric solution to the MSE loss with Nadaraya-Watson estimator, without retention.

All in all, as illustrated in Table 1, many existing methods can be unified under MIRAS.

## E   BEYOND EXISTING ATTENTIONAL BIASES AND RETENTION GATES

Here we provide the details of the alternative attentional biases and retention gates discussed in Section 4. We first propose several novel possible choices of attentional biases and then we discuss novel choices for retention gate.

### E.1   ALTERNATIVE ATTENTIONAL BIASES

**Variant 1: $\ell_p$-Attentional Bias.** As discussed in the main body, attentional bias defines the "similarity metric" and measures how well memory can recall the value, given its corresponding key. Although $\ell_2$ regression loss often is a natural choice, it is sensitive to noise in the data. A natural extension is to use $\ell_p$-norm class of objectives. That is, let $\mathcal{M}$ be the memory, $\boldsymbol{k}$ be the keys, and $\mathbf{v}$ be the values, we define $\ell_p$-attentional bias as:

$$\mathcal{L}(\mathcal{M}(W, \boldsymbol{k}_t); \mathbf{v}_t) = \|\mathcal{M}(\boldsymbol{k}_t) - \mathbf{v}_t\|_p^p, \tag{20}$$

where $p \in \mathbb{R}^{\geq 1}$ and $\|.\|_p$ is the $p$-norm. Although depending on the distribution of the data, we might want to use different values of $p$ (see Section 5), different values of $p$ can result in memory architectures with interesting properties. For the sake of simplicity, let memory be a matrix, i.e., $W \in \mathbb{R}^{m \times d}$ and $\mathcal{M}(W, \boldsymbol{k}_t) = W\boldsymbol{k}_t$, the closed form can be derived as:

$$W_t = W_{t-1} - \eta_t \nabla \ell(W_{t-1}; \boldsymbol{k}_t, \mathbf{v}_t) = W_{t-1} - p\,\eta_t\,\left(\text{Sign}(W_{t-1}\boldsymbol{k}_t - \mathbf{v}_t) \odot |W_{t-1}\boldsymbol{k}_t - \mathbf{v}_t|^{p-1}\right)\,\boldsymbol{k}_t^{\top}.$$

Let $p = 1$, the recurrence is simplified as:

$$W_t = W_{t-1} - \eta_t\,\text{Sign}(W_{t-1}\boldsymbol{k}_t - \mathbf{v}_t)\,\boldsymbol{k}_t^{\top}, \tag{21}$$

which means that the memory has only two values of $-1$ and $1$. We call this variation *value-less* associative memory, in which we store entities (keys) but map them into two extreme class of -1 and +1.

**Remark 1.** *One of the critical challenges to use the above update rule is in the backpropagation process, in which $\text{Sign}(\cdot)$ and $|\cdot|$ are non-differentiable and so might cause unstable training. To overcome this issue, we use $\text{Sign}(x) \approx \tanh(\alpha x)$, and $|x| = \sqrt{x^2 + \epsilon}$, as the smooth approximators of these functions.*

One simple interpretation for such behavior (i.e., value-less memory) is similar to the coping mechanism in humans (Loftus, 1993), in which the memory does not store the values for extreme events. This interpretation of protective memory in extreme events motivates our next variant.

**Variant 2: Huber Loss: Memory with Coping Mechanism.** While $\ell_2$-norm objective is a common choice for many statistical and machine learning tasks, it is known to be sensitive to outliers and extreme samples. This sensitivity extends to the use of $\ell_2$ loss for attentional bias. To address this and drawing motivation from robust regression literature, we suggest utilizing the Huber loss-type (Huber, 1992; Hastie et al., 2009) as the attentional bias, thereby reducing the negative impact of the outlier data on the memory learning process.

We can apply Huber-type loss in three different ways: The first approach is to define the summation of the Huber loss across different coordinates as the total loss, i.e.,

$$\ell(W; \boldsymbol{k}_t, \mathbf{v}_t) = \sum_j \mathcal{H}(\mathcal{M}(W, \boldsymbol{k}_t)_j - \mathbf{v}_{t,j}),$$

where $\mathcal{M}(W, \boldsymbol{k}_t)_j$ and $\mathbf{v}_{t,j}$ denote the $j$-th coordinate of $\mathcal{M}(W, \boldsymbol{k}_t)$ and $\mathbf{v}_t$ respectively. The function $\mathcal{H}(\cdot) : \mathbb{R} \mapsto \mathbb{R}$ is the Huber loss defined as

$$\mathcal{H}(a) = \begin{cases} \frac{1}{2}a^2 & \text{if } |a| \leq \delta \\ \delta\left(|a| - \frac{1}{2}\delta\right) & \text{if } |a| > \delta. \end{cases} \tag{22}$$

Utilizing this attentional bias can lead to various memory update rules. For example, for the matrix form memory $\mathcal{M}(W, \boldsymbol{k}_t) = W\boldsymbol{k}_t$, the update rule is given by

$$W_t = W_{t-1} - \eta_t \left[ \left( (W\boldsymbol{k}_t - \mathbf{v}_t)\boldsymbol{k}_t^T \right) \odot \left( \mathbf{I}(|W\boldsymbol{k}_t - \mathbf{v}_t| \leq \delta_t)\mathbf{1}^\top \right) \right.$$

$$\left. + \left( \delta_t \mathrm{Sign}(W\boldsymbol{k}_t - \mathbf{v}_t)\boldsymbol{k}^\top \right) \odot \left( \mathbf{I}(|W\boldsymbol{k}_t - \mathbf{v}_t| > \delta_t)\mathbf{1}^\top \right) \right] \tag{23}$$

In this formulation, the parameter $\delta_t$ decides the type of the memory used for each block of memory ($\ell_2$-norm objective or value-less) based on the context, making the memory more robust to outliers.

The second approach is to define the Huber-type loss based on the $\ell_2$ loss over all coordinates, i.e.,

$$\ell(W; \boldsymbol{k}_t, \mathbf{v}_t) = \mathcal{H}(\|\mathcal{M}(W, \boldsymbol{k}_t) - \mathbf{v}_t\|_2).$$

For simplicity of derivations, assume matrix memory $M(W, \boldsymbol{k}_t) = W\boldsymbol{k}_t$. Then using gradient descent for updating memory leads the memory update rule

$$W_t = W_{t-1} - \eta_t \begin{cases} (\mathcal{M}(W_{t-1}, \boldsymbol{k}_t) - \mathbf{v}_t)\,\boldsymbol{k}_t^T & \text{if} \quad \|\mathcal{M}(W_{t-1}, \boldsymbol{k}_t) - \mathbf{v}_t\|_2 \leq \delta_t, \\ \delta_t \frac{(\mathcal{M}(W_{t-1}, \boldsymbol{k}_t) - \mathbf{v}_t)}{\|\mathcal{M}(W_{t-1}, \boldsymbol{k}_t) - \mathbf{v}_t\|_2}\boldsymbol{k}_t^T & \text{Otherwise.} \end{cases} \tag{24}$$

Again, in the form equation 24, the parameter $\delta_t$ decides the type of the memory used ($\ell_2$-norm objective or normalized version) based on the context, making the memory more robust to outliers.

Finally, in the third approach, we present a smooth mixture method, in which the memory decides if for an incoming data it is better to use $\ell_2$ or $\ell_1$ attentional bias:

$$W_t = W_{t-1} - \begin{cases} \eta_t\, \nabla\ell_2(W_{t-1}; \boldsymbol{k}_t, \mathbf{v}_t) & \text{if} \quad \|\mathcal{M}(\boldsymbol{k}_t) - \mathbf{v}_t\| \leq \delta_t, \\ \eta_t\, \delta_t \nabla\ell_1(W_{t-1}; \boldsymbol{k}_t, \mathbf{v}_t) & \text{Otherwise.} \end{cases} \tag{25}$$

The role of parameter $\delta_t$ is the same as above.

**Variant 3: Memory Robust to Value Shifts.** Following the robustness requirement discussed in the previous section, we aim to design a memory mechanism that exhibits resilience against small shifts in the value parameter. A natural approach in this context is to employ a robust optimization formulation. Specifically, we define the loss function as the worst-case $\ell_2$ distance between the predicted memory output and the perturbed true value:

$$\mathcal{L}(\mathcal{M}(W, \boldsymbol{k}_t); \mathbf{v}_t) = \max_{\|\boldsymbol{\delta}\mathbf{v}_t\|_2 \leq \Delta} \frac{1}{2}\|\mathcal{M}(W, \boldsymbol{k}_t) - (\mathbf{v}_t + \boldsymbol{\delta}\mathbf{v}_t)\|_2^2. \tag{26}$$

This formulation seeks the memory parameters $W$ that perform well even under the adverse local perturbation of the true value $\mathbf{v}_t$ within an $\ell_2$ ball of radius $\Delta$. To solve the maximization problem in equation 26, we find the optimal perturbation $\boldsymbol{\delta}\mathbf{v}_t^*$. By solving this problem with respect to $\boldsymbol{\delta}\mathbf{v}_t$, we arrive at:

$$\boldsymbol{\delta}\mathbf{v}_t^* = \Delta\frac{-\mathcal{M}(W, \boldsymbol{k}_t) + \mathbf{v}_t}{\|\mathcal{M}(W, \boldsymbol{k}_t) - \mathbf{v}_t\|_2}$$

Substituting this optimal perturbation back into the loss function equation 26, we obtain the robust loss:

$$\mathcal{L}(\mathcal{M}(W, \boldsymbol{k}_t); \mathbf{v}_t) = \frac{1}{2}\|\mathcal{M}(W, \boldsymbol{k}_t) - \mathbf{v}_t\|_2^2 + \Delta\|\mathcal{M}(W, \boldsymbol{k}_t) - \mathbf{v}_t\|_2 + \frac{1}{2}\Delta^2.$$

This robust loss function is a combination of the standard $\ell_2$ loss and a term proportional to the $\ell_2$ norm of the error, scaled by the robustness parameter $\Delta$. The value of $\Delta$ thus controls the trade-off between fitting the nominal data and ensuring robustness against value perturbations.

For simplicity of the derivations, let us consider a constant value for $\Delta$, an Euclidean retention gate $\mathrm{Ret}_t(W, W_{t-1}) = \|W - W_{t-1}\|^2$, and an attentional bias term $\widetilde{\ell}(W; \mathbf{k}_t, \mathbf{v}_t) = \langle W - W_{t-1}, \nabla\ell(W_{t-1}; \mathbf{k}_t, \mathbf{v}_t) \rangle$. Furthermore, to simplify the memory operation, we assume a linear

matrix memory model $\mathcal{M}(W, \mathbf{k}_t) = W\mathbf{k}_t$. Under these assumptions, we can derive the memory update mechanism using gradient descent on the robust loss:

$$W_t = W_{t-1} - \eta \left( \left(\mathcal{M}(W_{t-1}, \boldsymbol{k}_t) - \mathbf{v}_t\right)\boldsymbol{k}_t^\top + \Delta \frac{\mathcal{M}(W_{t-1}, \boldsymbol{k}_t) - \mathbf{v}_t}{\|\mathcal{M}(W_{t-1}, \boldsymbol{k}_t) - \mathbf{v}_t\|_2} \boldsymbol{k}_t^\top \right)$$

In this update rule, the parameter $\Delta$, which governs the influence of the robustness term, can also be treated as a learnable parameter, allowing the model to adapt its robustness based on the observed data.

### E.2 ALTERNATIVE RETENTION GATES AND MEMORY STABILITY

**Variant 4: Memorization Over A Scaled Probability Simplex Via $f$-Divergence.** A common technique in learning to prevent numerical instabilities and exploding values is to restrict the search space to a bounded domain. Following this principle, to avoid numerical instabilities, we can constrained the variable $W_t$ to lie within a (scaled) probability simplex. In other words, we can restrict the state to lie in the constraint set

$$\mathcal{W} = \{W \mid \|W\|_1 = c \text{ and } W_{jl} \geq 0, \ \forall j, l\}.$$

In this set, each matrix $W$ can be viewed as a measure. Thus, in (Learning-Retaining Viewpoint) , we can utilize divergences over measures to define our premetric. For example, we can use $f$-divergence measure (Polyanskiy & Wu, 2025, Def 4.9), (Csiszar, 1967) to define $D_t(\cdot, \cdot)$. More specifically, let $f(\cdot)$ be a smooth strictly convex function from $\mathbb{R}^+$ to $\mathbb{R}$ with $f(1) = 0$. Then, we can define the $f-$ divergence between $W$ and $W'$ as

$$D_t(W, W') = \sum_{jl} W'_{jl} \, f\left(\frac{W_{jl}}{W'_{jl}}\right).$$

It is known that $f$-divergence is zero if and only if $W = W'$; see (Polyanskiy & Wu, 2025, Theorem 2.3). Using the above premetric as the retention gate and setting $\widetilde{\ell}(W; \boldsymbol{k}_t, \mathbf{v}_t) = \langle W - W_{t-1}, \nabla\ell(W_{t-1}; \boldsymbol{k}_t, \mathbf{v}_t)\rangle$ in (Learning-Retaining Viewpoint) , we get the update rule

$$W_t = W_{t-1} \odot g\left(-\zeta_t - \eta_t \nabla\ell(W_{t-1}; \boldsymbol{k}_t, \mathbf{v}_t)\right). \tag{27}$$

Here $g(\cdot)$ is the inverse of the mapping $f'$, i.e., $g(f'(\tau)) = \tau, \ \forall\tau$; the operator $\odot$ denotes the Hadamard (elementwise) product, and $\zeta_t$ should be chosen such that $\|W_t\|_1 = c$. Notice that since the function $f(\cdot)$ is strictly convex and smooth, its derivative is strictly increasing and hence $g(\cdot)$ is well defined. Conversely, for any strictly monotone function $g(\cdot)$, we can find its inverse function $g^{-1}$ (which is strictly increasing) and define $f(\tau) = \text{const} + \int_{\tau'=0}^\infty g^{-1}(\tau')d\tau'$. The term const should be chosen such that $f(1) = 0$. Then the update rule in equation 27 can be interpreted by the $f$-divergence regularization, as explained above. Therefore, one can directly choose a continuous monotonically increasing function $g(\cdot)$ and use equation 27 for memory update.

**Specializing to KL divergence.** Let us further make the above update rule explicit by using special function $f$. If we choose $f(\tau) = \tau \ln(\tau)$, then the $f$-divergence becomes the widely used KL divergence measure $D_t(W, W_{t-1}) = \sum_{jl} W_{jl} \log\left(\frac{W_{jl}}{(W_t)_{jl}}\right)$. In addition, we can also utilize the Shannon entropy as the global retention by regularizing deviations from uniform distribution, i.e., $G_t(W) = \sum_{jl} W_{jl} \log(W_{jl})$. Combining these choices of the local and global retention gates, we obtain the overall retention gate

$$\text{Ret}_t(W, W_{t-1}) = \frac{1}{\eta_t} \sum_{jl} W_{jl} \log\left(\frac{W_{jl}}{(W_t)_{jl}}\right) + \frac{1}{\alpha_t} \sum_{jl} W_{jl} \log(W_{jl})$$

Choosing the attentional bias $\widetilde{\ell}(W; \boldsymbol{k}_t, \mathbf{v}_t) = \langle W - W_{t-1}, \nabla\ell(W_{t-1}; \boldsymbol{k}_t, \mathbf{v}_t)\rangle$ and the above retention gate will lead to the update rule

$$W_t = \arg\min_W \langle W - W_{t-1}, \nabla\ell(W_{t-1}; \boldsymbol{k}_t, \mathbf{v}_t)\rangle + \frac{1}{\eta_t} \sum_{jl} W_{jl} \log\left(\frac{W_{jl}}{(W_t)_{jl}}\right) + \frac{1}{\alpha_t} \sum_{jl} W_{jl} \log(W_{jl}) \tag{28}$$

$$\text{s.t.} \quad \sum_{jl} W_{jl} = c, \ W_{jl} \geq 0, \ \forall jl \tag{29}$$

Attaching the Lagrange multiplier to the first constraint, the KKT conditions implies

$$(\nabla \ell(W_{t-1}; \boldsymbol{k}_t, \mathbf{v}_t))_{jl} + \left( \frac{1}{\eta_t} + \frac{1}{\alpha_t} \right) (1 + \log W_{jl}) - \frac{1}{\eta_t} \log ((W_{t-1})_{jl}) + \mu_t = 0, \quad \forall j, l \quad (30)$$

where $\mu_t$ should be chosen such that $\sum_{jl} W_{jl} = c$. Rearranging the terms and defining $\lambda_t = \frac{1/\alpha_t}{1/\alpha_t + 1/\eta_t}$, $\eta_t' = \frac{1}{1/\alpha_t + 1/\eta_t}$, we get the update rule

$$W_t \leftarrow c \, \text{Softmax} \left( (1 - \lambda_t) \log(W_{t-1}) - \eta_t' \nabla \ell(W_{t-1}; \boldsymbol{k}_t, \mathbf{v}_t) \right) \quad (31)$$

where $\lambda_t \in (0, 1)$ and $\eta' \in \mathbb{R}^+$ are the hyper-parameters that can be learned during training. The Softmax operator ensures that the output lies in the set $\mathcal{W}$.

Notice that the above calculation is done assuming the elements of $W$ form a probability vector (normalized to $c$). One can easily get update rules for the case that $W$ is a stochastic matrix (just by decomposing the problem across columns of $W$). In such a case keys and values can each be a probability vector and $W$ plays the stochastic matrix mapping keys to values. Moreover, we can even go beyond matrix memory and obtain update rules for other forms of parameters such as when $W$ is a neural network (or when the parameter $W$ is normalized per slice).

**Variant 5: Elastic Net Regularization: Hard and Soft Forgetting.** Elastic net is a powerful and popular tool in regression analysis to balance the feature selection capabilities of LASSO (Tibshirani, 1996) and bias reduction properties of Ridge regression (Hilt & Seegrist, 1977; Hoerl & Kennard, 1970). It has been widely used in different applications due to its ability to handle high-dimensional data and mitigate the effects of multicollinearity. Given this success, a natural question is what happens if we use this regularization scheme in our context.

Let us start based on (Learning-Retaining Viewpoint) to design our memorization scheme. As mentioned in (Learning-Retaining Viewpoint), the loss function $\widetilde{\ell}_t(W; \boldsymbol{k}_t, \mathbf{v}_t)$ is an approximation of the original function $\ell(\cdot)$, measuring our goodness-of-fit. Regularizing this loss with elastic net regularizer, we obtain the approximation

$$\widetilde{\ell}_t(W; \boldsymbol{k}_t, \mathbf{v}_t) = \langle W - W_{t-1}, \nabla \ell(W_{t-1}; \boldsymbol{k}_t, \mathbf{v}_t) \rangle.$$

with a global retention of $\mathrm{G}_t(W) = \frac{1}{2\beta} \|W\|_2^2 + \frac{1}{\alpha} \|W\|_1$. To fully specify the update rule of (Learning-Retaining Viewpoint), we also need to specify the premetric functions $\mathrm{D}_t(\cdot, \cdot)$. For the sake of keeping the update rule simple (and parallelizable), we can choose

$$\mathrm{D}_t(W, W_{t-1}) = \frac{1}{2} \|W - W_{t-1}\|_2^2.$$

These choices of the attentional bias and retention gate leads to the following update rule:

$$W_t = \mathcal{S}_\gamma \left( \lambda W_{t-1} - \zeta \nabla \ell(W_{t-1}; \boldsymbol{k}_t, \mathbf{v}_t) \right), \quad (32)$$

where $\gamma = \frac{\eta\beta}{\alpha(\eta+\beta)}$, $\lambda = \frac{\beta}{\beta+\eta}$, $\zeta = \eta\lambda$, and $\mathcal{S}_\gamma$ is the soft thresholding operator, applied element-wise. For each element, this operator is defined as

$$\mathcal{S}_\gamma(z) = \text{sign}(z) \max \{0, |z| - \gamma\}.$$

In other words, for large values of $z$, $\mathcal{S}_\gamma(z)$ makes $z$ closer to zero by $\gamma$ amount. If it is already in the $\gamma$-vicinity of zero, then it makes it zero (hard forget).

Equation equation 32 can be viewed as a combination of soft forgetting (obtained by multiplying $W$ by $\lambda \in (0, 1)$, and a hard forgetting (if it is smaller than $\gamma$). The hyperparameters $\gamma$, $\lambda$, and $\zeta$ can be learned. Notice that since the shrinkage operator is not differentiable, we can approximate it with its smooth approximation. For example, we can use $\mathcal{S}_\gamma(z) \approx \frac{|z| * \arctan(z/\gamma)}{\pi/2}$.

**Variant 6: Elastic Net Regularization: Forgetting via Soft-thresholding.** The elastic net regularizer can also be used in the (FTRL Viewpoint). In particular, in (FTRL Viewpoint), we can set

$$\frac{1}{\eta_t} R_t(W) = \frac{1}{\eta} \|W\|^2 + \frac{1}{\alpha} \|W\|_1$$

and use $\widehat{\ell}(W; x_i) = \langle W - W_{i-1}, \nabla\ell(W_{i-1}; x_i) \rangle$. Assuming initialization at $W_0 = 0$, these choices of attentional bias and retention gate leads to the update rules:

$$A_t = A_{t-1} - \eta\nabla\ell(W_{t-1}; \boldsymbol{k}_t, \mathbf{v}_t)$$
$$W_t = \mathcal{S}_{\eta/\alpha}(A_t) \tag{33}$$

Here $\mathcal{S}_{\eta/\alpha}(\cdot)$ is the soft-thresholding operator with parameter $\eta/\alpha$, which can be smoothly as explained in Variant 1.1.

**Variant 7: General $L_q$ Memory Stability.** Existing work is based on the retention gate choices $\mathrm{D}_t(W, W_{t-1}) = \|W - W_{t-1}\|_F^2$ or $R(W) = \|W\|_2^2$. However, one can choose other choices of retention gate. For example, in (FTRL Viewpoint), we can choose $L_q$ norm as the regularizer $R(W)$. More specifically, for $1 < q \le 2$, we can set

$$\frac{1}{\eta_t}R(W) = \frac{1}{2\eta(q-1)}\|W\|_q^2.$$

Using this retention gate and choosing $\widehat{\ell}_i(W; \boldsymbol{k}_t, \mathbf{v}_t) = \langle W - W_{i-1}, \nabla\ell(W_{i-1}; \boldsymbol{k}_t, \mathbf{v}_t) \rangle$ in (FTRL Viewpoint), leads to the update rule $W_t = -\eta\frac{A_t}{\|A_t\|_p^{p-2}}$, where $p = \frac{q}{q-1}$ and $A_t = \sum_{i=1}^t \nabla\ell(W_{i-1}; \boldsymbol{k}_t, \mathbf{v}_t)$; see (Shalev-Shwartz et al., 2012, Section 2.6). Here, $\odot$ denotes the Hadamard (element-wise) product and $|\cdot|$ is the element-wise absolute value operator. Assuming $W_0 = 0$, this update rule can be recursively written as:

$$A_t = A_{t-1} - \eta\nabla\ell(W_{i-1}; \boldsymbol{k}_t, \mathbf{v}_t), \quad \text{and} \quad W_t = \frac{A_t}{\|A_t\|_p^{p-2}}.$$

**Variant 8: Bregman Divergence as Retention Gate..** Another natural choice is to use Bregman divergence as retention gate, leading to a mirror descent-type algorithms. In particular, given a smooth strictly convex function $f(\cdot) : \mathbb{R} \mapsto \mathbb{R}$, we can define the function $F(W) = \sum_{jl} f(W_{jl})$. Based on this choice of function $F$, we define the Bregman divergence

$$D_t(W, W') = F(W) - F(W') - \langle W', W - W' \rangle$$

as our parametric function. Utilizing this retention gate and choosing $\widetilde{\ell}_t(W; \boldsymbol{k}_t, \mathbf{v}_t) = \langle W - W_{t-1}, \nabla\ell(W_{t-1}; \boldsymbol{k}_t, \mathbf{v}_t) \rangle$ in (Learning-Retaining Viewpoint), we obtain the update rule

$$W_t = g\left(-\eta\nabla\ell(W_{t-1}; \boldsymbol{k}_t, \mathbf{v}_t) + F'(W_{t-1})\right).$$

Here, $F'$ is the mapping obtained by applying $f'(\cdot)$ (the derivative of $f$) element-wise to all entries of its input matrix argument. The function $g$ is the inverse of the mapping $F'(\cdot)$, i.e., $g(F'(W)) = W$.

If we choose $f(\tau) = \frac{\tau^2}{2}$, then $F'(W)$ becomes the identity mapping and so is $g$. Therefore, the above update becomes simple gradient descent with no nonlinearity involved in the update rule. However, other choices of $f(\cdot)$ introduces additional nonlinearity in $g(\cdot)$, which can enhance the expressivity of our memory. For example, we can choose the function $f(\cdot)$ so that its derivative becomes the inverse sigmoid function, i.e., $f'(\tau) = \ln\left(\frac{\tau}{1-\tau}\right)$ with $f' : (0, 1) \mapsto \mathbb{R}$. Since $f'(\cdot)$ is strictly increasing, then the function $f(\cdot)$ (and hence $F(\cdot)$) is strictly convex. Therefore, the Bregman divergence is well defined. Moreover, the inverse of the function $f'(\cdot)$ becomes the sigmoid function, i.e., $g(\tau) = \sigma(\tau) = \frac{\exp(\tau)}{1+\exp(\tau)}$ with $g : \mathbb{R} \mapsto (0, 1)$. Then, the update of the memory becomes

$$W_t = \sigma\left(\ln\left(\frac{W_t}{1 - W_t}\right) - \eta\nabla\ell(W_{t-1}; \boldsymbol{k}_t, \mathbf{v}_t)\right),$$

where $\sigma$ is the sigmoid function operated element-wise on the entries of $W$, and the division operator $\frac{W_t}{1-W_t}$ is also performed element-wise. This update rule guarantees that the elements of $W_t$ remains within the interval $(0, 1)$.

# F   PARALLELIZABLE TRAINING AND EFFICIENT IMPLEMENTATION OF MIRAS' VARIANTS

While the design of MIRAS's variant are theoretically well-motivated, their recurrence is non-linear, potentially make their straightforward training slow for large scales. In this section, we build upon the work of Behrouz et al. (2025c); Sun et al. (2024) to make the training parallelizable. The main idea is to divide the sequence into chunks with size $b$ (usually is 16 or 64) and calculate the gradient for all tokens in the current chunk with respect to the last state of the memory in the previous chunk. That is, we use $\nabla\ell(\mathcal{M}_{t'}; \boldsymbol{k}_t, \mathbf{v}_t)$ instead of $\nabla\ell(\mathcal{M}_{t-1}; \boldsymbol{k}_t, \mathbf{v}_t)$, where $t'$ is the last state in the previous chunk.

Given the above trick, we can calculate all gradients at once and make the recurrence inside each chunk linear. However, to fully take advantage of accelerators, we need to reformulate the process as matrix multiplication. For MONETA, for the sake of clarity, assume $q = 2$. We follow the same algorithm as Behrouz et al. (2025c) and expand the recurrence as follows:

$$\mathcal{M}_t = \alpha_t \mathcal{M}_{t-1} - \eta_t \nabla\ell(\mathcal{M}_{t-1}; \boldsymbol{k}_i, \mathbf{v}_i)$$

$$= \beta_t \mathcal{M}_0 - \sum_{i=1}^{t} \eta_i \frac{\beta_t}{\beta_i} \nabla\ell(\mathcal{M}_{t'}; \boldsymbol{k}_i, \mathbf{v}_i), \tag{34}$$

where $t' = t - \mathrm{mod}(t, b)$, and $\beta_i = \prod_{j=1}^{i} \alpha_j$. For the sake of clarity, we focus on the first chunk, i.e., $t = b$ and so $t' = 0$, and explain the process for the case that $\mathcal{M}_t = W_t$ is linear. The process for 2-layer MLPs and other chunks is similar. Using $\ell_p$ loss function, we have:

$$\nabla\ell(W_0; \boldsymbol{k}_i, \mathbf{v}_i) = p\left(\mathrm{Sign}(W\boldsymbol{k}_t - \mathbf{v}_t) \odot |W\boldsymbol{k}_t - \mathbf{v}_t|^{p-1}\right) \boldsymbol{k}_t^\top$$

$$\Rightarrow \sum_{i=1}^{b} \eta_i \frac{\beta_b}{\beta_i} \nabla\ell(W_0; x_i) = p\mathbf{E}_b \odot \mathbf{B}_b \odot \mathrm{Sign}(W\boldsymbol{k}_t - \mathbf{v}_t) \odot (|W_0 K - V|^{p-1}) K^\top, \tag{35}$$

where $\mathbf{E}_b = [\eta_1 \quad \eta_2 \quad \ldots \quad \eta_b]$ and $\mathbf{B}_b$ is defined analogously on $\frac{\beta_b}{\beta_i}$s. For the sake of stablity in training, we use $\mathrm{Sign}(x) \approx \tanh(\alpha x)$ and $|x| = \sqrt{x^2 + \epsilon}$, where $\epsilon > 0$ is a small number (i.e., $\epsilon = 1e - 6$). As discussed before, the case that $q \neq 2$ appears as a normalization term on the memory. Similar to Titans (Behrouz et al., 2025c) and TTT (Sun et al., 2024), we do not apply this non-linearity inside each chunk and instead use it at the end of each chunk.

The process is the same for other two variants: (1) YAAD: We calculate the gradient of both $\ell_1$ and $\ell_2$ loss and use a masking based on $\|\mathcal{M}(\boldsymbol{k}_t) - \mathbf{v}_t\| \leq \delta_t$. (2) MEMORA: update has two non-linear part, i.e., softmax and log. As discussed above, we apply the softmax at the end of each chunk. Therefore, for the log function, we can calculate all the gradients of each chunk at first and then expand the recurrence with respect to the log of weights. Again, this process make the inside chunk recurrence linear and inter-chunk recurrence non-linear.

# G   EXPERIMENTAL SETUP

We perform experimental evaluation on the language modeling (Merity et al., 2017; Paperno et al., 2016), common-sense reasoning (Bisk et al., 2020; Zellers et al., 2019; Sakaguchi et al., 2021; Clark et al., 2018; 2019), and long context needle-in-haystack tasks (Hsieh et al., 2024). We compare our models with the state-of-the-art linear recurrent models, Transformers, and hybrid models (recurrent + attention). More specifically we compare with Transformer++ (Touvron et al., 2023), RetNet (Sun et al., 2023), Gated Linear Attention (GLA) (Yang et al., 2024b), Mamba (Gu & Dao, 2024), Mamba2 (Dao & Gu, 2024), DeltaNet (Yang et al., 2024c), TTT (Sun et al., 2024), and Gated DeltaNet (Yang et al., 2024a). By "Transformer++", we refer to a standard Transformer based on the LLaMa architecture (Touvron et al., 2023), incorporating modern practices like RMSNorm, SwiGLU, and RoPE.

We train our models with training context window of size 4096 using either FineWeb-Edu dataset (Penedo et al., 2024) (for LM and common-sense reasoning tasks) or C4 dataset (Raffel et al., 2020) (for scaling patterns). We use model sizes of 120M, 340M, 760M, and 1.3B parameters. We train small models (120M and 340M) on 15B tokens sampled from the dataset, the medium size model (760M) on 30B tokens, and the large model on 100B tokens.

Table 6: Architectural Details.

| Model | Block | Dim | Head | Peak LR | Token |
|-------|-------|------|------|---------|-------|
| 170M | 12 | 768 | 16 | 3e-3 | 15B |
| 350M | 24 | 1024 | 16 | 1.5e-3 | 15B |
| 780M | 24 | 1536 | 16 | 1.25e-3 | 30B |

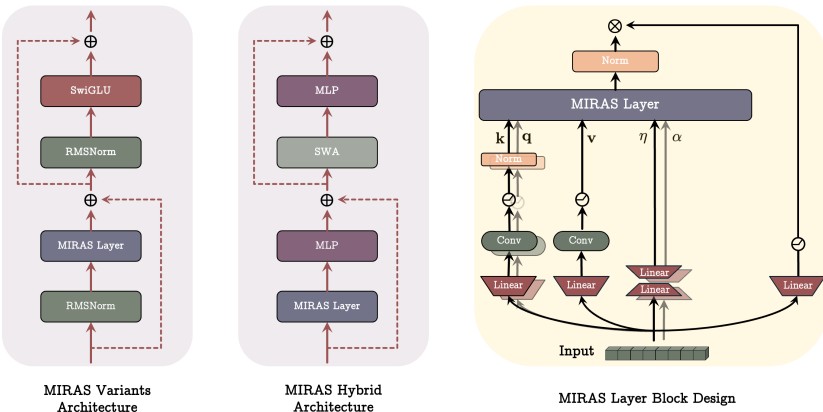

Figure 5: Visualization of the MIRAS' variant architecture, their hybrid counterpart with SWA, and block design of MIRAS layer.

## H ADDITIONAL EXPERIMENTAL RESULTS

### H.1 LANGUAGE MODELING

The full results for experiments on language modeling and common-sense reasoning tasks are reported in Table 7. Similar to 1.3B scale, our models achieve higher average accuracy compared to modern recurrent models.

### H.2 EFFICIENCY EVALUATIONS

In this section, we evaluate the training and inference throughput of MIRAS's variants with state-of-the-art sequence models, including Transformers. In particular, in 8K context window, the training throughput ($10^3$ T/s) of Transformers, Mamba, DeltaNet, and Titans are 48, 33, 39, and 37, respectively. MIRAS's variants of MEMORA, YAAD, and MONETA have training throughput of 34, 36, 37 ($10^3$ T/s), which is compatible and on par with state-of-the-art recurrent neural networks. It is notable that this throughput is achieved without any specially design kernel. Therefore, in summary: (1) Comparing to modern sequence models such as Mamba and DeltaNet (which also take advantage of optimized kernels), MIRAS's variants show competitive speed and are fast enough to be able to be scaled to larger scales; (2) Comparing to Titans, MIRAS's variants do not add significant computational overhead, despite they having more expressive attentional biases.

### H.3 MAD BENCHMARK

Next, we evaluate our models' performance and baselines' on MAD benchmark, which is a synthetic benchmark for evaluating the performance of sequence models in memorization, recall, compression, and copying tasks (Poli et al., 2024). The results are reported in Table 8. All MIRAS's variants achieve higher accuracy compared to baselines. Particularly in memorization, our models show relatively higher rate of improvements, which highlights the importance of going beyond conventional attentional biases.

### H.4 IN-CONTEXT RETRIEVAL TASK

In this section, we evaluate the performance of MIRAS's variants and baselines on in-context recall tasks, which is one of the most challenging benchmarks for recurrent neural networks. In this section, we follow Arora et al. (2024) and evaluate the models on SWDE (Lockard et al., 2019), NQ (Kwiatkowski et al., 2019), DROP (Dua et al., 2019), FDA (Arora et al., 2023), SQUAD (Rajpurkar et al., 2016), and TQA (Kembhavi et al., 2017). The results are reported in Table 9. Transformers still achieve the best results, outperforming all the recurrent models in in-context recall

Table 7: Performance of MIRAS' variants and recurrent- and Transformer-based baselines on language modeling and common-sense reasoning tasks. Hybrid models are marked with *. The best results are highlighted .

| Model | Wiki. ppl ↓ | LMB. ppl ↓ | LMB. acc ↑ | PIQA acc ↑ | Hella. acc_n ↑ | Wino. acc ↑ | ARC-e acc ↑ | ARC-c acc_n ↑ | SIQA acc ↑ | BoolQ acc ↑ | Avg. ↑ |
|---|---|---|---|---|---|---|---|---|---|---|---|
| | | | | | 340M params / 15B tokens | | | | | | |
| Transformer++ | 31.52 | 41.08 | 30.76 | 62.98 | 34.76 | 50.53 | 45.21 | 24.05 | 36.81 | 58.24 | 42.92 |
| RetNet | 32.50 | 49.73 | 28.24 | 62.61 | 34.15 | 50.91 | 44.27 | 23.62 | 36.79 | 59.72 | 42.54 |
| GLA | 28.51 | 43.02 | 28.73 | 64.05 | 35.96 | 50.00 | 54.19 | 24.29 | 37.13 | 58.39 | 44.09 |
| Mamba | 30.83 | 40.21 | 29.94 | 63.79 | 35.88 | 49.82 | 49.24 | 24.56 | 35.41 | 60.07 | 43.59 |
| DeltaNet | 28.65 | 47.30 | 28.43 | 63.52 | 35.95 | 49.63 | 52.68 | 25.37 | 37.96 | 58.79 | 44.04 |
| TTT | 27.44 | 34.19 | 30.06 | 63.97 | 35.71 | 50.08 | 53.01 | 26.11 | 37.32 | 59.83 | 44.51 |
| Gated DeltaNet | 27.01 | 30.94 | 34.11 | 63.08 | 38.12 | 51.60 | 55.28 | 26.77 | 34.89 | 59.54 | 45.42 |
| MONETA (ours) | 26.19 | 29.31 | 35.70 | 63.99 | 39.23 | 52.04 | 55.96 | 27.15 | 37.29 | 60.22 | 46.44 |
| YAAD (ours) | 26.61 | 29.11 | 34.09 | 64.93 | 39.86 | 51.12 | 54.75 | 28.64 | 33.82 | 60.29 | 45.93 |
| MEMORA (ours) | 27.16 | 30.44 | 33.68 | 65.21 | 39.17 | 51.23 | 53.40 | 27.99 | 34.1 | 59.29 | 45.51 |
| | | | | | 760M params / 30B tokens | | | | | | |
| Transformer++ | 25.21 | 27.64 | 35.78 | 66.92 | 42.19 | 51.95 | 60.38 | 32.46 | 39.51 | 60.37 | 48.69 |
| RetNet | 26.08 | 24.45 | 34.51 | 67.19 | 41.63 | 52.09 | 63.17 | 32.78 | 38.36 | 57.92 | 48.46 |
| Mamba2 | 22.94 | 28.37 | 33.54 | 67.90 | 42.71 | 49.77 | 63.48 | 31.09 | 40.06 | 58.15 | 48.34 |
| DeltaNet | 24.37 | 24.60 | 37.06 | 66.93 | 41.98 | 50.65 | 64.87 | 31.39 | 39.88 | 59.02 | 48.97 |
| TTT | 24.17 | 23.51 | 34.74 | 67.25 | 43.92 | 50.99 | 64.53 | 33.81 | 40.16 | 59.58 | 47.32 |
| Gated DeltaNet | 21.18 | 22.09 | 35.54 | 68.01 | 44.95 | 50.73 | 66.87 | 33.09 | 39.21 | 59.14 | 49.69 |
| Samba* | 20.63 | 22.71 | 39.72 | 69.19 | 47.35 | 52.01 | 66.92 | 33.20 | 38.98 | 61.24 | 51.08 |
| Gated DeltaNet-H2* | 19.88 | 20.83 | 39.18 | 68.95 | 48.22 | 52.57 | 67.01 | 35.49 | 39.39 | 61.11 | 51.49 |
| MONETA (ours) | 21.18 | 21.94 | 38.02 | 69.55 | 49.16 | 53.01 | 67.47 | 36.09 | 40.53 | 63.18 | 52.12 |
| YAAD (ours) | 20.99 | 21.57 | 37.85 | 69.14 | 50.02 | 53.93 | 67.78 | 36.27 | 41.01 | 63.34 | 53.98 |
| MEMORA (ours) | 22.28 | 22.31 | 38.19 | 67.82 | 49.30 | 53.28 | 63.57 | 36.15 | 40.94 | 62.96 | 51.52 |
| | | | | | 1.3B params / 100B tokens | | | | | | |
| Transformer++ | 18.53 | 18.32 | 42.60 | 70.02 | 50.23 | 53.51 | 68.83 | 35.10 | 40.66 | 57.09 | 52.25 |
| RetNet | 19.08 | 17.27 | 40.52 | 70.07 | 49.16 | 54.14 | 67.34 | 33.78 | 40.78 | 60.39 | 52.02 |
| Mamba2 | 16.56 | 12.56 | 45.66 | 71.87 | 55.67 | 55.24 | 72.47 | 37.88 | 40.20 | 60.13 | 54.89 |
| DeltaNet | 17.71 | 16.88 | 42.46 | 70.72 | 50.93 | 53.35 | 68.47 | 35.66 | 40.22 | 55.29 | 52.14 |
| Gated DeltaNet | 16.42 | 12.17 | 46.65 | 72.25 | 55.76 | 57.45 | 71.21 | 38.39 | 40.63 | 60.24 | 55.32 |
| Samba* | 16.13 | 13.29 | 44.94 | 70.94 | 53.42 | 55.56 | 68.81 | 36.17 | 39.96 | 62.11 | 54.00 |
| Gated DeltaNet-H2* | 15.91 | 12.55 | 48.76 | 72.19 | 56.88 | 57.77 | 71.33 | 39.07 | 41.91 | 61.55 | 56.18 |
| MONETA (ours) | 15.52 | 11.47 | 47.88 | 73.16 | 56.14 | 59.09 | 72.53 | 40.32 | 41.91 | 61.18 | 56.52 |
| YAAD (ours) | 15.18 | 11.89 | 47.23 | 72.81 | 56.46 | 59.02 | 72.14 | 40.05 | 40.73 | 61.86 | 56.39 |
| MEMORA (ours) | 15.90 | 12.04 | 48.67 | 73.10 | 55.99 | 57.36 | 71.55 | 37.92 | 40.19 | 61.34 | 55.87 |

Table 8: Performance of MIRAS' variants, and baselines on the synthetic benchmark of MAD (Poli et al., 2024). Our models achieve higher accuracy compared to all the baselines, including Transformers.

| | Compression | (Noisy) ICR | Fuzzy ICR | Selective Copying | Memorization | Average |
|---|---|---|---|---|---|---|
| Transformers | 49.4 | 100 | 48.2 | 95.9 | 83.8 | 75.46 |
| Gated DeltaNet | 44.8 | 100 | 32.5 | 96.2 | 81.7 | 71.04 |
| Titans | 49.6 | 100 | 49.7 | 99.4 | 83.5 | 76.44 |
| YAAD (ours) | 51.9 | 100 | 50.2 | 99.6 | 85.7 | 77.28 |
| MONETA (ours) | 51.1 | 100 | 48.9 | 99.6 | 85.4 | 77.00 |
| MEMORA (ours) | 50.5 | 100 | 48.7 | 99.6 | 85.1 | 76.78 |

tasks. Our variants of MIRAS, however, show competitive performance and improve the gap of recurrent models with Transformers.

Table 9: The performance of MIRAS' variants compared to baselines. While still Transformers achieve the best results in in-context recall tasks, our design of more expressive attentional bias can potentially reduce the performance gap with Transformers in future.

| | SWDE | NQ | DROP | FDA | SQUAD | TQA | Average |
|---|---|---|---|---|---|---|---|
| Transformers | 84.9 | 23.0 | 28.4 | 72.5 | 48.1 | 64.4 | 53.55 |
| Gated DeltaNet | 63.2 | 19.1 | 26.7 | 33.4 | 39.6 | 59.7 | 40.28 |
| Titans | 65.1 | 20.7 | 27.2 | 37.3 | 42.6 | 61.0 | 42.31 |
| YAAD (ours) | 66.2 | 20.9 | 27.2 | 38.1 | 42.7 | 61.3 | 42.73 |
| MEMORA (ours) | 65.5 | 20.5 | 26.9 | 38.2 | 43.0 | 61.2 | 42.55 |
| MONETA (ours) | 64.9 | 20.7 | 27.1 | 37.9 | 42.5 | 61.0 | 42.35 |

