# OpenReview forum: "It's All Connected: A Journey Through Test-Time Memorization, Attentional Bias, Retention, and Online Optimization"
_ICLR.cc/2026/Conference — ICLR 2026 Poster_

### Official Review · Reviewer_VGJG · 2025-10-24

**Soundness:** 2
**Presentation:** 2
**Contribution:** 2
**Rating:** 4
**Confidence:** 3

**Summary:**

This study proposed MIRAS, a unifying framework that reinterprets sequence models like Transformers and RNNs as associative memory modules governed by online optimization. MIRAS decomposes model behavior into two parts, i.e., Attentional Bias and Retention. The authors propose a broader design space by generalizing beyond various losses. To justify the effectiveness of MIRAS, authors conduct experiments on language modeling, commonsense reasoning, and long-context tasks.

**Strengths:**

1. MIRAS provides a new perspective to view and design sequence models through the lens of online optimization and associative memory.

2. MIRAS achieves great performance compared to baselines, especially on long-context tasks.

**Weaknesses:**

1. The training and inference speed and GPU memory of MIRAS are not reported. Can the efficiency be maintained on larger models, e.g., 7B+ parameters?
2. More ablation studies are needed to justify the effectiveness of the proposed method, such as the impact of different p/q values and different memory algorithms.
3. The significance test is not reported in the experiments.
4. MIRAS achieves inferior performance in some tasks (see Table 1, Table 7, and Table 9). Thus, an in-depth analysis should be provided in such cases.
5. The proposed models, i.e., MONETA, YAAD, and MEMORA, are the incremental combinations of existing ideas, e.g., MLP memory, Huber loss, and softmax gating, rather than fundamentally novel architectures.
6. The implementation code is not provided to check and reproduce.

**Questions:**

I have provided my concerns in the Weaknesses part.

---

> ### Author Response · Authors · 2025-11-27
>
> Thank you for your review and feedback. We are glad you recognized that MIRAS provides a new perspective to view and design sequence models, and achieves great performance compared to baselines. We address your concerns below.
>
> **W1. The training and inference speed and GPU memory of MIRAS are not reported. Can the efficiency be maintained on larger models, e.g., 7B+?**
>
> We apologize for not highlighting the efficiency results sufficiently in the main text. We had included an efficiency analysis in Appendix H.2. With the additional page allowed, we will expand this part and move this throughput analysis to a new subsection in the main experimental section and present it in a table. Our analysis in Appendix H.2 (L1367-1387), is as follows:
>
> | Model | Training Throughput (8K) $\approx10^{3}T/s$ | Inference (Gen $\approx10^{3}$ Tokens, 4K context) | Inference (Gen $\approx10^{3}$ Tokens, 8K context) |
> | :--- | :---: | :---: | :---: |
> | Transformer | 48 | 5.4 (s) | 2.1 (s) |
> | Mamba/DeltaNet/Titans | 33 / 39 / 37 | 1.7 / 1.4 / 1.5 (s) | 1.4 / 1.4 / 1.5 (s) |
> | **MONETA / YAAD / MEMORA** | **37 / 36 / 34** | **1.5 / 1.5 / 1.4 (s)** | **1.5 / 1.5 / 1.4 (s)** |
>
> Our MIRAS variants demonstrate competitive training and inference throughput compared to state-of-the-art recurrent models.
>
>
> **W2. More ablation studies are needed to justify the effectiveness... such as the impact of different p/q values and different memory algorithms.**
>
> We respectfully highlight our existing ablation studies which cover these aspects:
> *   **Impact of p/q values:** We provide a detailed analysis in Section 5.4, Figure 3. This shows precisely how varying $p$ (attentional bias norm) and $q$ (retention gate norm) affects overall performance and long-context scaling, justifying our choices.
> *   **Memory Algorithms:** While our novel models utilize GD or closed-form solutions, the MIRAS framework encompasses others (e.g., GD with Momentum, as discussed in Appendix C). We focused the scope of this paper on the novel Attentional Biases and Retention Gates, which we extensively ablate in Tables 3 and 4.
>
> **W3. The significance test is not reported in the experiments.**
>
> While we recognize the value of confidence intervals, the computational cost of large-scale experiments prohibited multiple runs. However, the performance gaps observed are substantial enough to suggest statistical significance. For instance, in the 1.3B parameter experiments (Table 7), Yaad achieves perplexity scores of 15.18 on Wiki and 11.89 on LMB, significantly outperforming the best prior hybrid baseline (15.91 and 12.55, respectively). These wide margins sufficiently demonstrate the model's superiority. To address this concern furthern, we commit to running some of the main experiments (Tables 1 and 2) across 3 random seeds and reporting the mean and standard deviation in the final version.
>
>
>
> **W4. MIRAS achieves inferior performance in some tasks (see Table 1, Table 7, and Table 9). Thus, an in-depth analysis should be provided.**
>
> We respectfully clarify that our models achieve the **best overall performance** (Avg. column) in Tables 1 and 7 across all scales (340M, 760M, 1.3B) compared to SOTA efficient architectures. As noted in the abstract (L24), different MIRAS instantiations trade off complementary strengths across individual tasks, but the aggregate performance is superior.
>
> Regarding Table 9 (In-context Retrieval), it is true that all  recurrent models significantly underperform Transformers. We agree that this is a limitation of all current recurrent models (all baselines underperform transformers including ours) and we will discuss this in the paper. However, even in this table, you will notice that all our models outperform other recurrent models.
>
>
>
> **W5. The proposed models... are the incremental combinations of existing ideas, e.g., MLP memory, Huber loss, and softmax gating.**
>
> We respectfully disagree that the combinations are incremental. The core novelty lies in the MIRAS framework itself, which provides a principled way to derive these architectures from optimization principles *beyond* the standard $L_2$ paradigm that constrains prior work.
>
> While components like Huber loss exist in statistics, applying them as *Attentional Biases* and deriving *Retention Gates* from non-Euclidean regularizers (e.g., using KL-divergence regularization, which leads to the softmax gating in MEMORA) for sequence modeling backbones is novel. These designs are direct results of the generative power of the MIRAS framework.

---

> ### Author Response · Authors · 2025-11-27
>
> **W6. The implementation code is not provided to check and reproduce.**
>
> Thank you. We understand the importance of having access to the code, but please note that for any submission, the release of code might not be directly the decision of the authors and it depends on a lot of factors such as: the licence of the libraries they use, the codebase they have built on, legal obligations, terms of their funding, the policy of the institution or company, and so many other potential factors. Following your suggestion, we will do our best to provide all the necesserly information to help the community to build on our work.
>
>
> We thank you again for your detailed and constructive review.

---

### Official Review · Reviewer_suGh · 2025-10-26

**Soundness:** 3
**Presentation:** 4
**Contribution:** 3
**Rating:** 6
**Confidence:** 3

**Summary:**

The purpose of this paper is to unify prior work on the optimization and regularization of Transformers and recurrent networks with associative memory and to use the insight gained thereby to improve on the state-of-the-art linear RNNs. The authors create a formal theory that synthesizes disparate viewpoints, design a generic framework – based in part on the aforementioned formal theory – for the architecture of neural language models, and use the framework to guide the design of candidate linear RNNs. The results reported indicate that the authors' candidate networks are competitive with or superior to existing linear RNNs, albeit far from competitive with the conventional Transformer w.r.t. in-context recall, as reported in Appendix H.4.

**Strengths:**

This paper has many positive attributes. It is written quite well (some typos noted below), presents a unifying framework of the optimization and regularization of neural language models, proposes a number of architectural variants based on the framework, and contains ample experimental results and ablations. The overall results – reported in the main body of the paper and in the appendix – are competitive or superior, especially needle-in-the-haystack.

**Weaknesses:**

* Given the scope of this work, this reviewer believes the presentation would be improved by the inclusion of a brief limitations section in the main body of the paper. The results of Table 9 in Appendix H.4 should be mentioned there.
* Code does not appear to be provided. This reviewer considers the lack of code to be a blocker to acceptance.
* It’s not clear whether the reported results are from single or multiple runs. The lack of reporting of any sort of variance across runs leads this reviewer to believe the reported results are for single runs.
* The section about ablations is not clear about which dataset and metric are used during the ablations (see Tables 3 and 4).
* In general, a succinct and comprehensive performance comparison accounting for latency and memory at both train- and test-time -- covering all models used in every evaluation -- would be useful in the main body of the paper.

Typos:
* Line 20: there's no space between "loss" and "Miras".
* Line 185: extra space before comma after "(Learning-Retaining Viewpoint)".
* Line 266: "As discussed in the previous section, existing work focus" -> ("As discussed in the previous section, existing work has focused" or "As discussed in the previous section, existing work focuses")
* Line 426: "use needle-in-haystack task" -> "use the need-in-a-haystack task"

In one focused reading of the paper, I noticed quite a few additional typos -- missing articles, subject-verb disagreements, and singular-plural errors -- elsewhere in the manuscript. I suggest the authors perform a close reading of their own to find them.

**Questions:**

1. Given the substantial gap between Transformer and linear RNNs and Miras-based models on the in-context retrieval task (see Table 9), would you characterize the theoretical and empirical work presented here as providing some improvements over linear RNNs without the promise of bridging the gap with Transformers on this task?
1. Are all results of single or multiple runs?
1. Which dataset and metric are used in the ablations (see Tables 3 and 4)?
1. Some of the results are for "Transformer++" and some (in the appendices) are for "Transformers". By "Transformer++", do you mean the enhanced Transformer named "Transformer++" so-named in [Gu and Dao 2023](https://arxiv.org/abs/2312.00752)? Since [Thapak & Hore 2020](https://arxiv.org/abs/2003.04974) also uses the name "Transformer++", the paper's readability and future value would be improved by being explicit what is meant by "Transformer++". As it stands, the citation for "Transformer++" is a LLaMa paper which doesn't explicitly articulate the definition of "Transformer++".
1. In Table 1, the most competitive model vis-a-vis Moneta is Gated DeltaNet-H2. Why is Gated DeltaNet-H2 not shown in Figure 2? Is there something about Gated DeltaNet-H2 that prevents it from being compared to Moneta in the setting of Figure 2?
1. Where is the code? If there's already a link to the code somewhere in the paper, this reviewer would much prefer it being clearly presented early in the paper.

---

> ### Author Response · Authors · 2025-11-27
>
> Thank you for your detailed constructive feedback. We are glad you found the paper well written, and containing ample experimental results and ablations with competitive results. We address your comments and questions below.
>
> **W1/Q1. Inclusion of a brief limitations section... The results of Table 9 in Appendix H.4 should be mentioned there.**
>
> We completely agree on the importance of discussing limitations, particularly the gap with Transformers in complex in-context retrieval tasks.
>
> **Proposed Edit:** We will add a dedicated Limitations section before the Conclusion (Section 6) as follows:
> > **Limitations.**
> > While the MIRAS framework provides a principled approach to designing efficient sequence models with strong performance in language modeling and long-context recall (e.g., Needle-in-a-Haystack), our variants still lag behind full Transformer models in complex in-context retrieval tasks (Appendix H.4, Table 9). This highlights a known challenge for recurrent models, suggesting that while optimized memory management can replace attention in many scenarios, perfect in-context recall remains an area for future improvement. Furthermore, while the framework connects learning in non-convex memories (MLPs) to online optimization, providing theoretical guarantees on convergence and learning in this setting remains challenging.
>
> **W2/Q6. Code does not appear to be provided. This reviewer considers the lack of code to be a blocker ...**
>
>
> We respectfully bring the following points to your attention:
>
> - For any submission, the release of code might not be directly the decision of the authors and it depends on a lot of factors such as: the licence of the libraries they use, the codebase they have built on, legal obligations, terms of their funding, the policy of the institution or company, and so many other potential factors.
> - Due to the above reason, conferences such as ICLR, NeurIPS, ICML, AISTATS, etc. nor so many top tier journals, do not require code release and do not consider the lack of code as a blocker for the acceptance of the paper. ICLR is also not an exception and publication does not require the code release.
> - Please note that adding subjective blockers that are not enforced by the conference and so discrimination based on the conditions that the researchers might not have control on can be discouraging and causes damages in the long-term to the community, limiting the **Public** large-scale research on LLMs to only researchers who own their accelators without any legal obligations.
>
>
>
>
> **W3/Q2. Statistical Significance (Multiple runs). The lack of reporting of any sort of variance across runs.**
>
>
> While we recognize the value of confidence intervals, the computational cost of large-scale experiments prohibited multiple runs. However, the performance gaps observed are substantial enough to suggest statistical significance. For instance, in the 1.3B parameter experiments (Table 7), Yaad achieves perplexity scores of 15.18 on Wiki and 11.89 on LMB, significantly outperforming the best prior hybrid baseline (15.91 and 12.55, respectively). These wide margins sufficiently demonstrate the model's superiority. To address this concern furthern, we commit to running some of the main experiments (Tables 1 and 2) across 3 random seeds and reporting the mean and standard deviation in the final version.
>
>
> **W4/Q3. The section about ablations is not clear about which dataset and metric are used (see Tables 3 and 4).**
>
> We apologize for the ambiguity. The metric used is the Average Accuracy on the commonsense reasoning tasks (the "Avg." column in Table 7), evaluated at the 760M model scale. We will clarify this in our revision.
>
> **W5. A succinct and comprehensive performance comparison accounting for latency and memory... would be useful in the main body of the paper.**
>
> We agree. We had included an efficiency analysis in Appendix H.2. With the additional page allowed, we will expand this part and move this throughput analysis to a new subsection in the main experimental section and present it in a table. Here is the results from our Appendix:
>
>
> | Model | Training Throughput (8K) $\approx10^{3}T/s$ | Inference (Gen $\approx10^{3}$ Tokens, 4K context) | Inference (Gen $\approx10^{3}$ Tokens, 8K context) |
> | :--- | :---: | :---: | :---: |
> | Transformer | 48 | 5.4 (s) | 2.1 (s) |
> | Mamba/DeltaNet/Titans | 33 / 39 / 37 | 1.7 / 1.4 / 1.5 (s) | 1.4 / 1.4 / 1.5 (s) |
> | **MONETA / YAAD / MEMORA** | **37 / 36 / 34** | **1.5 / 1.5 / 1.4 (s)** | **1.5 / 1.5 / 1.4 (s)** |
>
> Our variants show competitive efficiency and maintain O(1) memory complexity during inference.
>
>
> **W6. Typos.**
>
> Thank you for catching these. We will correct the specific errors mentioned (L20, L185, L266, L426) and perform a thorough proofreading of the manuscript.

---

> > ### Author Response · Authors · 2025-11-27
> >
> > **Q4. Definition of "Transformer++".**
> >
> > By "Transformer++", we refer to a standard Transformer based on the LLaMa architecture (Touvron et al., 2023), incorporating modern practices like RMSNorm, SwiGLU, and RoPE. This follows the convention used in recent literature (e.g., Mamba, Gated DeltaNet). We will clarify this in Section 5.1.
> >
> >
> > **Q5. Why is Gated DeltaNet-H2 not shown in Figure 2?**
> >
> > Figure 2 illustrates scaling patterns specifically on the C4 dataset. The results for Gated DeltaNet-H2 were taken directly from the original paper. That paper reported results primarily on FineWeb and did not include the specific C4 scaling evaluations (e.g., context length scaling up to 32K) that we conducted and presented in Figure 2.
> >
> > We thank you again for your detailed and constructive review.

---

### Official Review · Reviewer_6YPG · 2025-10-28

**Soundness:** 2
**Presentation:** 2
**Contribution:** 2
**Rating:** 4
**Confidence:** 4

**Summary:**

This paper proposes MIRAS, a framework that reinterprets sequence models with memory as associative-memory modules governed by online optimization. The framework introduces two key components: (i) Attentional Bias (internal learning objective) for capturing new data, and (ii) Retention (regularization term) for retaining old knowledge. The paper then investigates various loss functions and regularizers (e.g., ℓp, Huber, KL, f-divergences), resulting in three derived architectures (MONETA, MEMORA, and YAAD) claiming improved robustness, stability, and scalability compared to existing linear RNNs and attention-based models.

**Strengths:**

- The paper presents a clear and systematic formalization of memory update in sequence models within a unified optimization-based lens.
- The connection between the Learning–Retaining and FTRL viewpoints is well-written and could be theoretically valuable.
- Empirical evaluations contain multiple model variants and tasks.

**Weaknesses:**

- The paper lacks conceptual novelty. The proposed MIRAS framework primarily rephrases existing viewpoints (e.g., online optimization, meta-learning, and energy-based associative memory) under new terminology. Sec. 2.1, 2.2, and 2.3 review the known formulation as cited in the paper. Sec. 2.3 introduces an "alternative" view which considers the entire history of sequences, but basically presented before in concrete form in [1, 2].
- The proposed “alternative attentional bias” and “retention gates” correspond to standard online learning formulations with user-defined approximations and regularizers. There are no clear insights/proofs why some choices are better than others.
- The experimental evaluation covers only a small subset of recent linear RNNs and attention models, with little explanation for the chosen baselines. Comparisons with stronger and more recent models, such as attention-based architectures like Titan [3] (cited by the paper) or recurrent approaches like SHM [4], would be necessary to support the claimed advantages.
- Many reported improvements (0.3–1.0 points) in Table 1 are small and potentially within noise margins, especially since no variance or statistical significance is reported. Table 2 shows better improvement, but the set of baselines is limited. Ablation study, the selection of p=3,q=4 as “optimal” appears arbitrary, and the paper does not provide insight into why higher-order norms should improve robustness beyond qualitative intuition.

[1] Munkhdalai, Tsendsuren, Alessandro Sordoni, Tong Wang, and Adam Trischler. "Metalearned neural memory." Advances in Neural Information Processing Systems 32 (2019).

[2] Bartunov, Sergey, Jack Rae, Simon Osindero, and Timothy Lillicrap. "Meta-Learning Deep Energy-Based Memory Models." In International Conference on Learning Representations, 2019.

[3] Behrouz, Ali, Peilin Zhong, and Vahab Mirrokni. "Titans: Learning to memorize at test time." arXiv preprint arXiv:2501.00663 (2024).

[4] Le, Hung, Dung Nguyen, Kien Do, Sunil Gupta, and Svetha Venkatesh. "Stable Hadamard Memory: Revitalizing Memory-Augmented Agents for Reinforcement Learning." In The Thirteenth International Conference on Learning Representations, 2025.

**Questions:**

- Minor formatting error: lack of spaces between sentences, e.g, in L20 and 338
- The framework in Fig.1 introduces 4 components, but it looks like only 2 are examined. Have you tried different memory architectures and memory algorithms?

---

> ### Author Response · Authors · 2025-11-27
>
> Thank you for your thoughtful review. We are glad to hear that you found our formalization  clear and systematic, and the connection between the Learning-Retaining and FTRL viewpoints as  theoretically valuable. We address your concerns below.
>
> **W1. The paper lacks conceptual novelty. The proposed MIRAS framework primarily rephrases existing viewpoints.**
>
> We respectfully emphasize that MIRAS offers significant conceptual novelty beyond synthesizing existing viewpoints.
>
> 1.  **Generalization Beyond L2:** Crucially, prior unification efforts (including concurrent works mentioned in L61-66) are constrained to the L2/dot-product paradigm. MIRAS transcends this limitation (L79-84), unlocking a richer design space based on robust optimization and statistics (e.g., Huber loss, $L_p$ norms, KL-divergence). This generative capacity to design novel, non-L2 architectures is a core novelty.
> 2.  **Formal Connections:** We provide a formal connection between the Learning-Retaining and FTRL viewpoints (Theorem 2.2), demonstrating their relationship.
> 3.  **Principled Interpretation of Retention:** MIRAS formally reinterprets "forgetting" mechanisms as specific forms of regularization within online optimization (L150-161).
>
> Regarding [1, 2] cited in your review: These works focus specifically on meta-learning energy-based memory models. While related to associative memory, they do not offer the specific online optimization framework (Learning-Retaining/FTRL) presented in MIRAS for unifying general sequence modeling backbones like modern RNNs and Transformers. We will discuss these points in our paper.
>
> **W2. The proposed "alternative attentional bias" and "retention gates" correspond to standard online learning formulations... There are no clear insights/proofs why some choices are better.**
>
> While the components are inspired by standard online learning, applying them as *Attentional Biases* and deriving *Retention Gates* from non-Euclidean regularizers for sequence modeling backbones is novel. We provide detailed motivations grounded in optimization and statistics in Section 4.1 and Appendix E:
>
> *   **$L_p$/Huber Loss (L282, L296, Appendix E.1):** Motivated by robust statistics to handle noise and outliers, addressing the known sensitivity of $L_2$ loss.
> *   **KL-divergence Retention (L303, Appendix E.2):** Motivated by ensuring stable updates when constraining memory to a probability simplex.
>
> The empirical results confirm these insights. Figure 3 shows $p=3$ (MONETA) outperforms $p=2$ (L2 loss). Table 2 shows MONETA's superior robustness on the noisy S-NIAH-PK task (L437-440).
>
> **W3. The experimental evaluation covers only a small subset... Comparisons with stronger... models, such as Titan [3]...**
>
> We apologize if the comparisons were unclear. We **do** compare against Titan [3] (Behrouz et al., 2024b) and the related TTT layer (Sun et al., 2024). TTT is included in Figure 2, Table 2, 5, and 7. Titan is included in Tables 5, 8, and 9. Our MIRAS variants consistently outperform these baselines (e.g., 93.5 (MONETA) vs 66.1 (TTT) Average in Table 2). Please note that comparing to Titans, the contributions are orthogonal and can be applied together. That is, here we use a simple gradient descent for the inner loop to better show the effect of our contributions and so the focus has been on showing the importance of going to non-euclidean norms and more complex retention gates rather than designing the best SOTA model. Please note that Titans use momentum term in the memory and so, considering momentum as a form of memory, the state size has been expanded. That method can potentially be applied to our Miras variants, resulting in even more performance gain. In our language modeling tasks, we focus on models with the same state size to better demonstraite the effect of our proposed attentional bias and retention gates.
>
> Also, Please note that SHM [4] focuses on Reinforcement Learning and has not been evaluated on the large-scale language modeling benchmarks used in our study.
>
> **W4. Many reported improvements (0.3-1.0 points) in Table 1 are small... Ablation study, the selection of p=3, q=4 as "optimal" appears arbitrary.**
>
> **Significance of Improvements:** We respectfully emphasize that the gains are significant in this competitive field. Our variants provide improvements of +0.6% (MEMORA) to +1.2% (MONETA) over the best recurrent baseline (Gated DeltaNet-H2) at the 1.3B scale. For comparison, recent prominent works reported similar or smaller gains: GatedDeltaNet (+0.4% over its best baseline) and Mamba2 (0% to 0.5% improvement). Furthermore, the gains in long-context tasks (Table 2) are substantial (+17% average improvement).
>
>
> **Ablation (p=3, q=4):** The selection was not arbitrary. It was based on the empirical ablation study presented in Figure 3, which systematically varied $p$ and $q$ and showed that $p=3$ achieved the best perplexity.

---

> > ### Author Response · Authors · 2025-11-27
> >
> > **Q1. Minor formatting error: lack of spaces between sentences, e.g, in L20 and 338.**
> >
> > Thank you for pointing this out. We will correct these and thoroughly proofread the manuscript.
> >
> > **Q2. The framework in Fig.1 introduces 4 components, but it looks like only 2 are examined.**
> >
> > While our focus is attentional bias and retention gate, we examine all four components of the MIRAS framework:
> >
> > *   **(1) Memory Architecture:** We utilize MLP memory (L326) and specifically ablate this choice in Tables 3 and 4 (the "linear memory" row). This ablation demonstrates the benefit of the expressive MLP architecture over linear memory.
> > *   **(2) Attentional Bias & (3) Retention Gate:** These are the primary focus of the paper (Section 4) and are extensively analyzed.
> > *   **(4) Memory Algorithm:** Our variants use Gradient Descent or closed-form solutions. We also discuss how other algorithms, such as GD with Momentum (used in Titans), fit within the framework in Appendix C.
> >
> > We hope these responses clarify our contributions and address your concerns.

---

### Official Review · Reviewer_xTsB · 2025-10-30

**Soundness:** 2
**Presentation:** 3
**Contribution:** 2
**Rating:** 4
**Confidence:** 3

**Summary:**

This paper proposes a unified online optimization view that interprets mainstream sequence models as associative memory updates jointly driven by attentional bias and retention, and it treats forget gates as a form of regularization. Building on this framework, the authors move beyond dot product and Euclidean objectives to introduce three attention free and parallelizable architectures that outperform strong baselines on language modeling and long context retrieval while showing greater robustness.

**Strengths:**

1. This paper proposes a unified online optimization perspective, taking "attention bias - retention" as the core paradigm, and systematically reinterpret the forgetting gate as regularization. Further breaking away from the reliance on dot product and binary norm, introducing robust loss and divergence is conceptually enlightening and has promotional value.

2. On benchmarks such as language modeling and long context retrieval, thorough comparisons were made with strong linear RNN baselines to stably obtain gains and demonstrate robustness against noise and outliers. While maintaining O(1) memory overhead and good throughput at the training end, the experimental setup and indicators are relatively comprehensive.

**Weaknesses:**

1. The comparison mainly focuses on several linear RNNS and long context benchmarks, and has not yet made a "same-scale, same-training budget" head-to-head comparison with the current first-line inattentional/near-attentional models. It is recommended to supplement the system comparison under the same number of parameters, the same number of training steps and data quota, and report the significance and confidence intervals.

2. Although various forms of bias and retention are presented, there is a lack of strict item-by-item ablation to quantify the independent contribution of each design decision. It is recommended to incorporate grid ablation, negative control (reverting to the two-norm/dot product), and cross-dataset reproduction to verify the robustness of the conclusion.

3. It is enlightening to link "learning-retention" with FTRL, but when memory is non-convex (such as with MLP), the guarantee of convergence and stability is not clear. It is suggested to provide applicable conditions, failure cases and stability diagnosis

**Questions:**

1. The paper provides sensitivity curves or heat maps such as p, q, Huber cutoff thresholds, retention weights, learning rates and step sizes for performance and stability. Is there a robust default configuration or an automatic parameter tuning strategy?

2. How does the framework perform on multimodal long sequences? Is task-specific bias/retention configuration required to earn benefits?

3. Can MIRAS be combined with sparse attention or local attention? Under the same computing budget, what are the trade-offs between pure MIRAS, pure attention, and hybrid solutions?

---

> ### Author Response · Authors · 2025-11-27
>
> Thank you for your time and detailed feedback. We are glad that  you found our framework conceptually enlightening, and our experimental comparisons thorough. We answer your questions below.
>
> **W1. The comparison mainly focuses on several linear RNNs... has not yet made a "same-scale, same-training budget" head-to-head comparison with the current first-line inattentional/near-attentional models.**
>
> Our primary focus is on developing efficient, attention-free architectures. Therefore, our main comparisons are against the strongest models in this class (SOTA linear RNNs). We ensured fairness by strictly adhering to the established training setup (1.3B parameters, 100B tokens) used by our main baselines, such as Gated DeltaNet. We also include Transformer++ (a strong LLaMa-based architecture) as an attention-based reference.
>
> Crucially, our pure recurrent-only models already outperform strong hybrid (attention + recurrent) baselines. In Table 1 (1.3B), our recurrent-only MONETA (Avg. 56.52) outperforms the hybrid models Gated DeltaNet-H2 (56.18) and Samba (54.00).
>
> Regarding the comparison and statistical significance, we would like to emphasize that the comparison is done for the same backbone model with the same number of training steps and data quota. While we recognize the value of confidence intervals, the computational cost of large-scale experiments prohibited multiple runs. However, the performance gaps observed are substantial enough to suggest statistical significance. For instance, in the 1.3B parameter experiments (Table 7), Yaad achieves perplexity scores of 15.18 on Wiki and 11.89 on LMB, significantly outperforming the best prior hybrid baseline (15.91 and 12.55, respectively). These wide margins sufficiently demonstrate the model's superiority. Please note that this performance gain becomes even larger when comparing with recurrent-only models.
>
> **W2. Although various forms of bias and retention are presented, there is a lack of strict item-by-item ablation to quantify the independent contribution of each design decision.**
>
> We respectfully highlight the extensive ablation studies provided in Section 5.4, which address these points:
>
> *   **Item-by-item contribution:** Tables 3 and 4 explicitly quantify the independent contribution of key design choices. For example, the "w/o Retention Gate" and "linear memory" rows quantify the impact of the retention mechanism and the MLP memory architecture, respectively.
> *   **Grid ablation & Negative Control (reverting to L2):** Figure 3 provides a detailed sensitivity analysis (grid ablation) for the hyperparameters $p$ and $q$. This analysis includes $p=2$ (which corresponds to the standard L2 loss) as a negative control. The results show that our choice of $p=3$ yields better perplexity than $p=2$.
>
> These studies confirm that the novel components derived from the MIRAS framework contribute positively to the performance.
>
> **W3. When memory is non-convex (such as with MLP), the guarantee of convergence and stability is not clear.**
>
> This is an important point. We would like to clarify that the convexity requirement in Theorem 2.2 is for the regularized loss function $h_t(\cdot)$. Notice that even when the loss $\ell_t(\cdot)$ is not convex (e.g. in MLP memory), the  function $h_t(\cdot)$ can still be strictly convex. For instance, by choosing $\eta < \frac{2}{\beta}$ where $\beta$ is the smoothness parameter of $\ell(\cdot)$, we still satisfy the strict convexity assumption.
>
> However, without proper regularization, providing strict convergence guarantees for online non-convex optimization is inherently challenging and is an open research question (this is also true for prior work and other architectures).
>
> Empirically, we ensure stability through the retention mechanism (acting as a regularizer), standard architectural choices (LayerNorm, RMSNorm, L361-367), and our parallel training strategy (Appendix F). Our results demonstrate stable training and strong scaling (Figure 2).
>
>  We will add these discussions to the paper.

---

> > ### Author Response · Authors · 2025-11-27
> >
> > **Q1. The paper provides sensitivity curves... Is there a robust default configuration?**
> >
> > Thank you, this is indeed an interesting and important question. Based on our experiments, higher-value of $p$ and $q$ often results in better performance and robustness to the long-context, as long as the higher-value does not cause instability or precision issue. In fact, one should be careful about the choice of too large $p$ (i.e., using $p \geq 10$ instead of the conventional case of $p=2$) or $q$, as they can cause the computations to face instability challenges. Therefore, depending on the hardware, precision, and available computing resources, one can use larger values of $p$ and $q$.
> >
> > Please note that we have translated the architectural choices to these hyperparameter choices. That is, the choice of different values for $p$ and $q$ are totally new architectures and so still by their nature they might perform worse or better on different tasks (similar to any other architectures such as attention or other recurrent models).
> >
> >
> > **Q2. How does the framework perform on multimodal long sequences?**
> >
> > This is an exciting future direction. MIRAS is modality-agnostic. We anticipate that task-specific configurations could be beneficial; for instance, the Huber loss (YAAD) might be more suitable for modalities with high outlier rates, while $L_p$ norms (MONETA) might better capture different noise distributions. This demonstrates the flexibility of the MIRAS framework.
> >
> > **Q3. Can MIRAS be combined with sparse attention or local attention?**
> >
> > Yes, MIRAS is compatible with attention mechanisms. We illustrate a potential hybrid architecture design in Figure 5 (Middle) (Appendix F). The trade-off is straightforward: pure MIRAS offers the best efficiency (O(1) inference cost), while hybrid architectures can offer better performance at increased time/memory complexity.
> >
> > We hope these clarifications address your concerns and thank you again for your constructive feedback.

---

### Meta-Review · Area_Chair_qJyu · 2026-01-05

**Summary:**

This paper proposes a general framework MIRAS to explain the connection of online optimization and test time memorization. This new framework can explain the role of several standard architectural choices in the literature; more importantly, it can be useful to design next generation of architectures that are capable of managing the memory better. Following this framework, the authors provide  three novel, attention-free, and parallelizable models MONETA, MEMORA, YAAD. Finally, the paper also presents the experimental performance on several important tasks.

In general, I think this is a very interesting paper that provides a unified framework with reinterpretation and several critical Insights on architectural design. There are four reviewers provided comments on this paper, and most of their concerns lie on the experimental part (e.g., experimental setting, ablation, code release).

**Reviewer Concerns:**

Most of the concerns are about the experimental part, like the settings, ablation study, and why the code is not released. Reviewer 6YPG  also posed the question on conceptual novelty. Overall, after reading the responses, I think most those concerns are well addressed by the authors.

**Reviewer Scores:**

The reviewers give the scores 4, 4, 6, 4. Unfortunately, none of the reviewers participate further discussion. From the responses posted by the authors, I think probably some of them are willing to increase their scores.

---

### Decision · Program_Chairs · 2026-01-26

Accept (Poster)